# Evolutionary conservation of centriole rotational asymmetry in the human centrosome

**Noémie Gaudin[1], Paula Martin Gil[1], Meriem Boumendjel[1], Dmitry Ershov[2,3], Catherine Pioche-Durieu[1], Manon Bouix[1], Quentin Delobelle[1], Lucia Maniscalco[1], Than Bich Ngan Phan[1], Vincent Heyer[4,5,6,7], Bernardo Reina-San-Martin[4,5,6,7], Juliette Azimzadeh[1]\***

[1]Université Paris Cité, CNRS, Institut Jacques Monod, Paris, France; [2]Image Analysis Hub, C2RT, Institut Pasteur, Paris, France; [3]Hub de Bioinformatique et Biostatistique – Département Biologie Computationnelle, Institut Pasteur, Paris, France; [4]Institut de Génétique et de Biologie Moléculaire et Cellulaire (IGBMC), Ilkirch, France; [5]Institut National de la Santé et de la Recherche Médicale (INSERM), Illkirch, France; [6]Centre National de la Recherche Scientifique (CNRS), Illkirch, France; [7]Université de Strasbourg, Illkirch, France

**\*For correspondence:**
juliette.azimzadeh@cnrs.fr

**Competing interest:** The authors declare that no competing interests exist.

**Abstract** Centrioles are formed by microtubule triplets in a ninefold symmetric arrangement. In flagellated protists and animal multiciliated cells, accessory structures tethered to specific triplets render the centrioles rotationally asymmetric, a property that is key to cytoskeletal and cellular organization in these contexts. In contrast, centrioles within the centrosome of animal cells display no conspicuous rotational asymmetry. Here, we uncover rotationally asymmetric molecular features in human centrioles. Using ultrastructure expansion microscopy, we show that LRRCC1, the ortholog of a protein originally characterized in flagellate green algae, associates preferentially to two consecutive triplets in the distal lumen of human centrioles. LRRCC1 partially co-localizes and affects the recruitment of another distal component, C2CD3, which also has an asymmetric localization pattern in the centriole lumen. Together, LRRCC1 and C2CD3 delineate a structure reminiscent of a filamentous density observed by electron microscopy in flagellates, termed the 'acorn.' Functionally, the depletion of LRRCC1 in human cells induced defects in centriole structure, ciliary assembly, and ciliary signaling, supporting that LRRCC1 cooperates with C2CD3 to organizing the distal region of centrioles. Since a mutation in the *LRRCC1* gene has been identified in Joubert syndrome patients, this finding is relevant in the context of human ciliopathies. Taken together, our results demonstrate that rotational asymmetry is an ancient property of centrioles that is broadly conserved in human cells. Our work also reveals that asymmetrically localized proteins are key for primary ciliogenesis and ciliary signaling in human cells.

## Editor's evaluation

This work shows that, contrary to a widely accepted view, centrioles of the human centrosome are rotationally asymmetric, a feature previously known only from centrioles in flagellated protists and multiciliated cells. The authors identify LRRCC1, implicated in ciliary disease, as an asymmetrically localized protein of the centriole lumen and show that it contributes to proper centriole structure, ciliary assembly, and ciliary signaling.

## Introduction

Centrioles are cylindrical structures with a characteristic ninefold symmetry, which results from the arrangement of their constituent microtubule triplets (*LeGuennec et al., 2021*). In animal cells, centrioles are essential for the assembly of centrosomes and cilia. The centrosome, composed of two centrioles embedded in a pericentriolar material (PCM), is a major organizer of the microtubule cytoskeleton. In addition, most vertebrate cells possess a primary cilium, a sensory organelle that assembles from the oldest centriole within the centrosome, called mother centriole (*Kumar and Reiter, 2021*).

Centrioles within the centrosome show no apparent rotational asymmetry, that is, no structural asymmetry of the microtubule triplets. In vertebrates, the mother centriole carries distal appendages (DAs) and subdistal appendages arranged in a symmetric manner around the centriole cylinder (*Kumar and Reiter, 2021*). In contrast, the centriole/basal body complex of flagellates, to which the animal centrosome is evolutionary related, is characterized by marked rotational asymmetries (*Azimzadeh, 2021*; *Yubuki and Leander, 2013*). In flagellates, an array of fibers and microtubules anchored asymmetrically at centrioles controls the spatial organization of the cell (*Feldman et al., 2007*; *Yubuki and Leander, 2013*). The asymmetric attachment of cytoskeletal elements appears to rely on molecular differences between microtubule triplets. In the green alga *Chlamydomonas reinhardtii,* Vfl1p (variable flagella number 1 protein) localizes principally at two triplets near the attachment site of a striated fiber connecting the centrioles (*Silflow et al., 2001*). This fiber is absent or mispositioned in the *vfl1* mutant, leading to defects in centriole position and number, and overall cytoskeleton disorganization (*Adams et al., 1985*; *Feldman et al., 2007*). In the same region, a rotationally asymmetric structure termed the 'acorn' was observed in the centriole lumen by transmission electron microscopy. The acorn appears as a filament connecting five successive triplets and is in part colocalized with Vfl1p (*Geimer and Melkonian, 2005*; *Geimer and Melkonian, 2004*).

We recently established that Vfl1p function is conserved in the multiciliated cells (MCCs) of planarian flatworms, which was recently confirmed in *Xenopus* (*Basquin et al., 2019*; *Nommick et al., 2022*). MCCs assemble large numbers of centrioles that are polarized in the plane of the plasma membrane to enable the directional beating of cilia (*Meunier and Azimzadeh, 2016*), like in *C. reinhardtii*. The planarian ortholog of Vfl1p is required for the assembly of two appendages that decorate MCC centrioles asymmetrically, the basal foot and the ciliary rootlet (*Basquin et al., 2019*). Depleting Vfl1p orthologs in planarian or xenopus MCCs alters centriole rotational polarity, reminiscent of the *vfl1* phenotype in *C. reinhardtii* (*Adams et al., 1985*; *Basquin et al., 2019*; *Nommick et al., 2022*). Intriguingly, the human ortholog of Vfl1p, called LRRCC1 (leucine rich repeat and coiled coil containing 1) localizes at the centrosome despite the lack of rotationally asymmetric appendage in this organelle (*Andersen et al., 2003*; *Muto et al., 2008*). Furthermore, a homozygous mutation in the *LRRCC1* gene was identified in two siblings affected by a ciliopathy called Joubert syndrome (JBTS), suggesting that LRRCC1 might somehow affect the function of non-motile cilia (*Shaheen et al., 2016*).

Here, we show that LRRCC1 localizes in a rotationally asymmetric manner in the centrioles of the human centrosome. We further establish that LRRCC1 is required for proper ciliary assembly and signaling, which likely explains its implication in JBTS. LRRCC1 affects the recruitment at centrioles of another ciliopathy protein called C2CD3 (C2 domain containing 3), which we found to also localize in a rotationally asymmetric manner, forming a pattern partly reminiscent of the acorn described in flagellates. Our findings uncover the unanticipated rotational asymmetry of centrioles in the human centrosome and show that this property is connected to the assembly and function of primary cilia.

## Results

### LRRCC1 localizes asymmetrically at the distal end of centrioles

To investigate a potential role of LRRCC1 at the centrosome, we first sought to determine its precise localization. We raised antibodies against two different fragments within the long C-terminal coiled-coil domain of LRRCC1 (Ab1, 2), which both stained the centrosome region in human retinal pigmented epithelial (RPE1) cells (*Figure 1a*, *Figure 1—figure supplement 1a*), as previously reported (*Muto et al., 2008*). Labeling intensity was decreased in LRRCC1-depleted cells for both antibodies, supporting their specificity (Figure 4a and b, *Figure 1—figure supplement 1*). LRRCC1 punctate labeling in the centrosomal region indicated that it is present within centriolar satellites, confirming a previous finding that LRRCC1 interacts with the satellite component PCM1 (*Gupta et al.,*

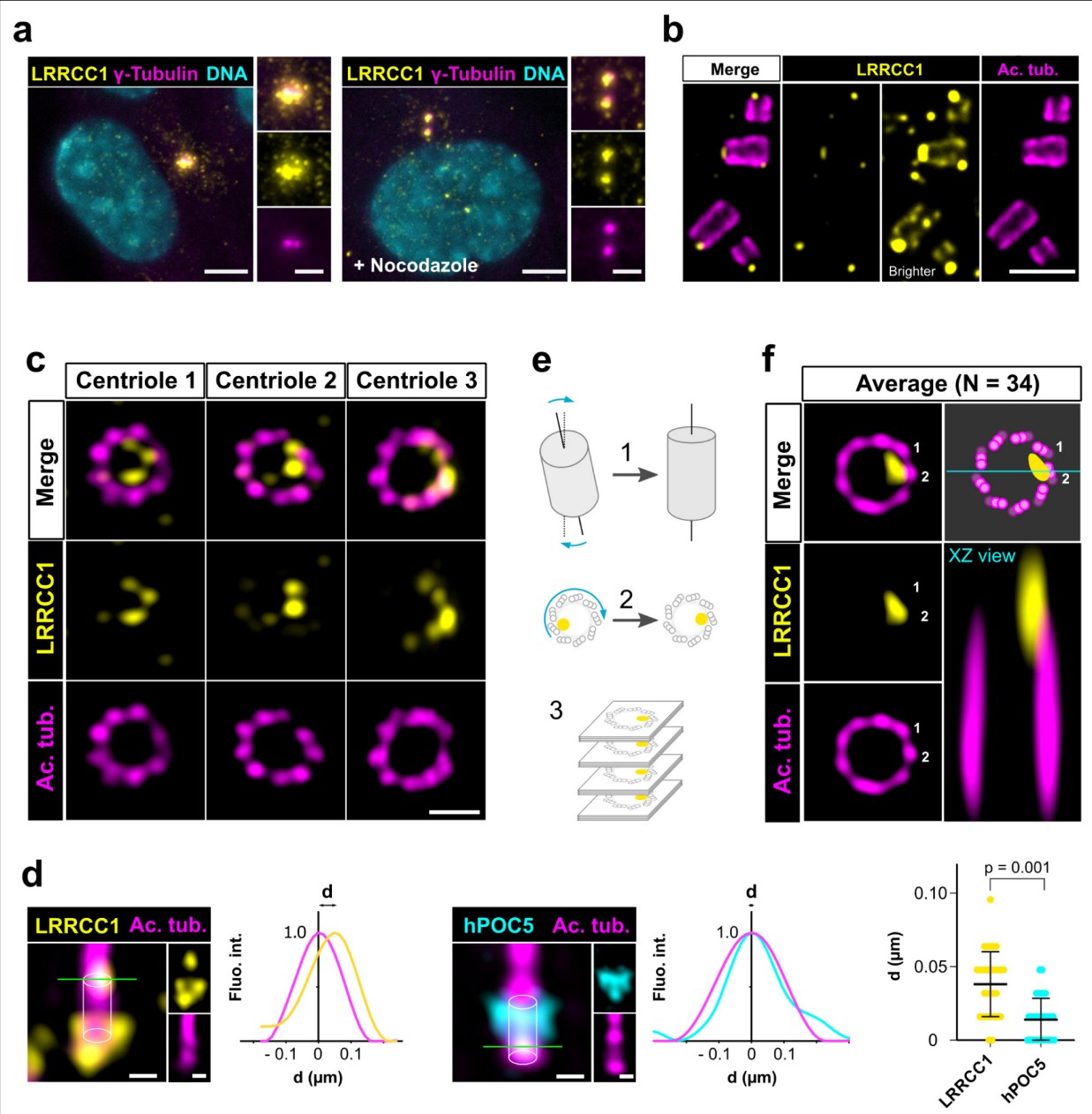

**Figure 1.** LRRCC1 is localized in a rotationally asymmetric manner at the distal end of centrioles in the human centrosome. (**a**) LRRCC1 localization in non-treated RPE1 cells (left) or in cells treated with nocodazole to disperse the pericentriolar satellites (right). LRRCC1 (Ab2, yellow), γ-tubulin (PCM, magenta), and DNA (cyan). Bar, 5 μm (insets, 2 μm). (**b**) Longitudinal view of centrioles and procentrioles in the duplicating centrosome of an RPE1 cell analyzed by ultrastructure expansion microscopy (U-ExM). LRRCC1 (Ab2, yellow), acetylated tubulin (magenta). Bar, 0.5 μm. (**c**) Centrioles from WT RPE1 cells as seen from the distal end. LRRCC1 (Ab2, yellow), acetylated tubulin (magenta). Images are maximum intensity projections of individual z-sections encompassing the LRRCC1 signal. Note that an apparent shift between channels occurs when centrioles are slightly angled with respect to the imaging axis. Bar, 0.2 μm. (**d**) Lateral distance between LRRCC1 (left, yellow) or hPOC5 (middle, cyan) signal intensity peaks and the centriole center (given by the position of acetylated tubulin intensity peak, magenta) in ciliated RPE1 cells. Individual intensity profiles were measured along the green lines. The approximate position of the centriole is shown (white cylinders). Note that LRRCC1 and hPOC5 were also detected at the periphery of the centriole, towards the proximal end for LRRCC1 and in the appendage region for hPOC5. Bar, 0.2 μm. Right: interpeak distance (**d**). Bars, mean ± SD, 31 cells from two different experiments (Kolmogorov–Smirnov test). (**e**) Workflow for calculating the average staining from 3D-reconstructed individual centrioles generated from confocal z-stacks. The brightest part of LRRCC1 signal was used as a reference point to align the centrioles. (**f**) Average LRRCC1 staining obtained from 34 individual centrioles viewed from the distal end, in transverse and longitudinal views. A diagram representing the average pattern in transverse view is also shown.

The online version of this article includes the following source data and figure supplement(s) for figure 1:

*Figure 1 continued on next page*

*Figure 1 continued*

**Figure supplement 1.** Characterization of LRRCC1 expression in CRISPR or RNAi-treated cells.

**Figure supplement 1—source data 1.** Western blot analysis of LRRCC1 in *Figure 1—figure supplement 1b*.

**Figure supplement 2.** Pipeline for generating average protein maps.

*2015*). After nocodazole depolymerization of microtubules to disperse satellites, a fraction of LRRCC1 was retained at centrioles (*Figure 1a*, *Figure 1—figure supplement 1a*), providing evidence that LRRCC1 is also a core component of centrioles. To determine LRRCC1 localization more precisely within the centriolar structure, we used ultrastructure expansion microscopy (U-ExM) (*Gambarotto et al., 2019*) combined with imaging on a Zeiss Airyscan 2 confocal microscope, thereby increasing the resolution by a factor of ~8 compared to conventional confocal microscopy. We found that LRRCC1 localizes at the distal end of centrioles as well as of procentrioles (*Figure 1b*). Strikingly, and unlike other known centrosome components, LRRCC1 decorated the distal end of centrioles in a rotationally asymmetric manner. Indeed, LRRCC1 was detected close to the triplet blades and towards the lumen of the centriole (*Figure 1c*). The staining was often associated with two or more consecutive triplets, one of them being usually more brightly labeled than the others. In addition, a fainter staining was consistently detected along the entire length of all triplets (*Figure 1b*, brighter exposure). This pattern was observed in both RPE1 and HEK 293 cells and was obtained with both anti-LRRCC1 antibodies (*Figure 1—figure supplement 1h*), supporting its specificity. We verified that LRRCC1 asymmetric localization was also observed in unexpanded cells by directly analyzing immunofluorescence samples by Airyscan microscopy (*Figure 1d*). We measured the lateral distribution of LRRCC1 signal intensity peak relative to the long axis of the centriole. The distance between peaks was greater for LRRCC1 than for hPOC5, a marker that localizes symmetrically in the centriole (*Azimzadeh et al., 2009*; *Le Guennec et al., 2020*; *Schweizer et al., 2021*), confirming the asymmetry of LRRCC1 staining. The distal pattern obtained by U-ExM showed some variability, especially in the distance between LRRCC1 and the centriole wall (*Figure 1c*), which could result from the fact that centrioles were not perfectly orthogonal to the imaging plan. To obtain a more accurate picture of LRRCC1 localization, we generated 3D reconstructions that we realigned, first along the vertical axis, then with respect to one another using the most intense region of the LRRCC1 labeling as a reference point (*Figure 1e*, *Figure 1—figure supplement 2a and b*). An average 3D reconstruction was then generated (*Figure 1f*) and revealed that LRRCC1 was mainly associated to one triplet, and to a lesser extent to its direct neighbor counterclockwise, on their luminal side. A longitudinal view confirmed that LRRCC1 is principally located at the distal end of centrioles.

Together, our results show that LRRCC1 is localized asymmetrically within the distal centriole lumen, establishing that centrioles within the human centrosome are rotationally asymmetric.

## The localization pattern of LRRCC1 is similar at the centrosome and in mouse MCCs

LRRCC1 orthologs are required for establishing centriole rotational polarity in planarian and xenopus MCCs, like in *C. reinhardtii* (*Basquin et al., 2019*; *Nommick et al., 2022*; *Silflow et al., 2001*). It is therefore plausible that LRRCC1-related proteins localize asymmetrically in MCC centrioles, and indeed, Lrrcc1 was recently found associated to the ciliary rootlet in xenopus MCCs (*Nommick et al., 2022*). To determine whether LRRCC1 also localizes at the distal end of MCC centrioles in addition to its rootlet localization, and if so, whether LRRCC1 localization pattern resembles that observed at the centrosome, we analyzed mouse ependymal and tracheal cells by U-ExM. In in vitro differentiated ependymal cells, the labeling generated by the anti-LRRCC1 antibody was consistent with our observations in human culture cells. Mouse Lrrcc1 localized asymmetrically at the distal end of centrioles, opposite to the side where the basal foot is attached (*Figure 2a*), as determined by co-staining with the basal foot marker γ-tubulin (*Clare et al., 2014*). Lrrcc1 was also present at the distal end of procentrioles forming via either the centriolar or acentriolar pathways (i.e., around parent centrioles or deuterosomes, respectively) (*Figure 2b*). We also examined tracheal explants, in which centrioles were docked and polarized at the apical membrane in higher proportions (*Figure 2c*). We obtained an average image of Lrrcc1 labeling from 35 individual centrioles aligned using the position of the basal foot as a reference point. This revealed that Lrrcc1 is principally located in the vicinity of three triplets opposite to the basal foot,

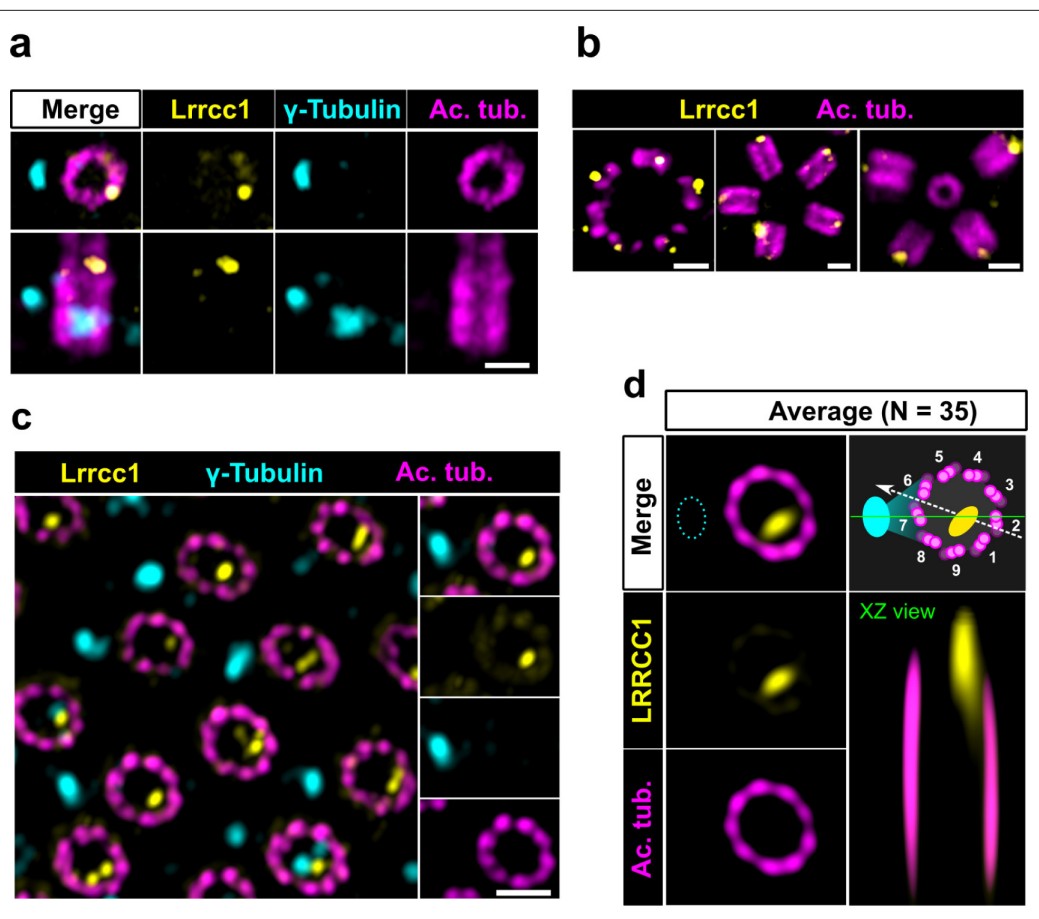

**Figure 2.** The LRRCC1 rotationally asymmetric pattern is conserved in mouse multiciliated cells (MCCs). (**a**) Centrioles in the cytoplasm of mouse ependymal cells differentiating in vitro analyzed by ultrastructure expansion microscopy (U-ExM), in longitudinal and transverse view. Lrrcc1 (Ab2, yellow), γ-tubulin (basal foot cap, cyan), and acetylated tubulin (magenta). Of note, γ-tubulin was also detected in the proximal lumen of centrioles. Bar, 0.2 μm. (**b**) Procentrioles assembling via the centriolar (right) or the deuterosome pathway (left and center) in ependymal cells. Lrrcc1 (Ab2, yellow), acetylated tubulin (magenta). Bar, 0.2 μm. (**c**) Transverse view of centrioles docked at the apical membrane in fully differentiated mouse tracheal cells, viewed from the distal end. Lrrcc1 (Ab2, yellow), γ-tubulin (cyan), and acetylated tubulin (magenta). Bar, 0.2 μm. (**d**) Average image generated from 35 individual centrioles from mouse trachea, viewed from the distal end, shown in transverse and longitudinal views. The position of the basal foot (cyan dotted line) stained with γ-tubulin was used as a reference point to align the centrioles. A diagram of the average pattern in transverse view is shown, in which the direction of ciliary beat (*Schneiter et al., 2021*) is represented by a dotted arrow and the basal foot axis by a green line. Triplets are numbered counterclockwise from the LRRCC1 signal.

to the right of basal foot main axis (triplet numbers 9, 1, and 2 on the diagram in *Figure 2d*). Lrrcc1 was located farther away from the triplet wall than in centrioles of the centrosome, but this was likely an effect of a deformation of the centrioles (*Figure 2c and d*) caused by the incomplete expansion of the underlying cartilage layer in tracheal explants. In agreement, Lrrcc1 was close to the triplets in ependymal cell monolayers, which expand isometrically. Besides the distal centriole staining, we found no evidence that Lrrcc1 is associated to the ciliary rootlet in mouse MCCs, unlike in xenopus. The Lrrcc1 pattern in mouse MCCs was thus similar to the pattern observed at the human centrosome.

Together, these results show that LRRCC1 asymmetric localization is a conserved feature of mammalian centrioles, presumably linked to the control of centriole rotational polarity and ciliary beat direction in MCCs.

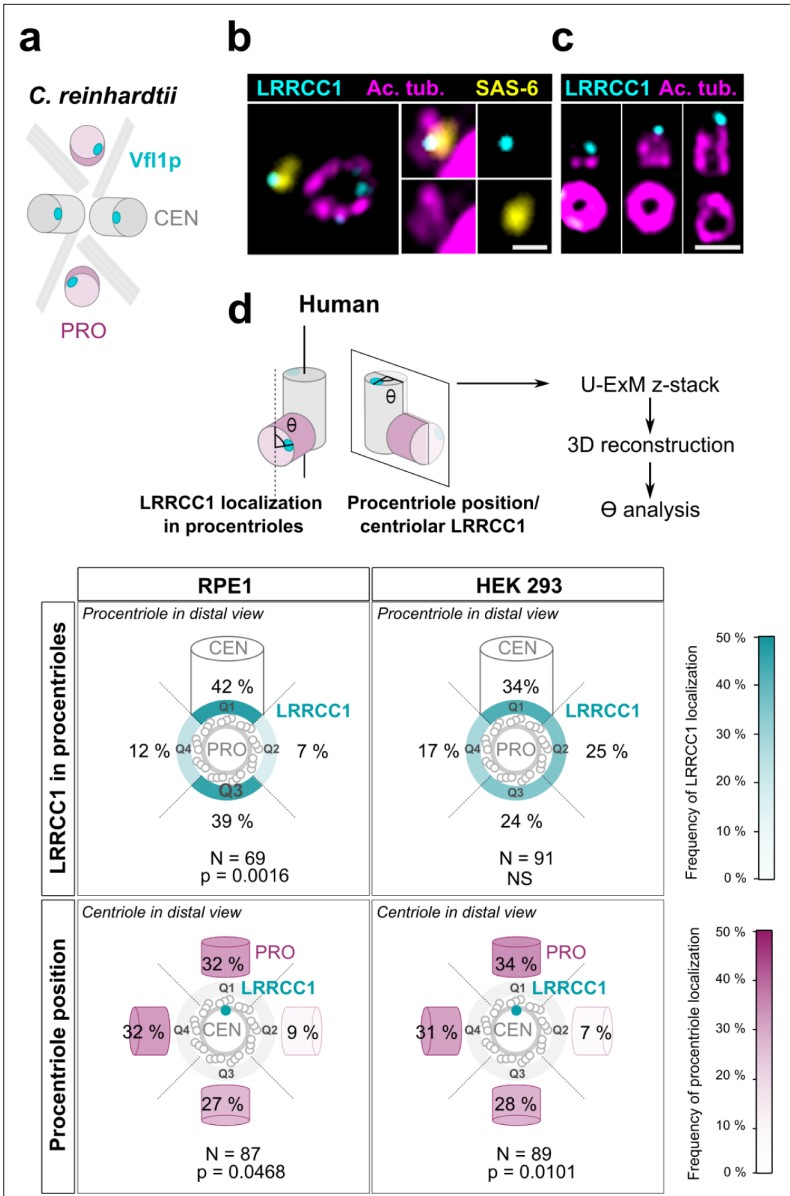

**Figure 3.** Procentriole assembly site is partly correlated with centriole rotational polarity. (**a**) Diagram showing the localization of Vfl1p (cyan) in the centrioles/basal bodies (gray) and procentrioles/probasal bodies (pink) of *C. reinhardtii*. The microtubule roots are also shown. (**b**) Early stage of procentriole assembly stained for LRRCC1 (Ab2, cyan), SAS-6 (yellow), and acetylated tubulin (magenta) in a HEK 293 cell. The brightness of the acetylated tubulin labeling was increased in the insets. Bar, 0.1 µm. (**c**) Successive stages of centriole elongation in HEK 293 cells stained for LRRCC1 (Ab2, cyan) and acetylated tubulin (magenta). Bar, 0.1 µm. (**d**) Location of LRRCC1 in the procentrioles (top panels) and position of the procentriole relative to its parent centriole polarity (bottom panels), in RPE1 and HEK 293 centrioles analyzed by ultrastructure expansion microscopy (U-ExM). For each diplosome, the angle between LRRCC1 in the procentriole and the centriole long axis (top panels), or between the procentriole and LRRCC1 in the centriole (bottom panels) was measured. The number of diplosomes analyzed is indicated. p-Values are indicated when statistically different from a random distribution ($\chi^2$ test).

## Procentriole assembly site is partly correlated with centriole rotational polarity

In *C. reinhardtii*, cytoskeleton organization and flagellar beat direction depend on the position and orientation at which new centrioles arise during cell division. Reflecting the stereotypical organization of centrioles and procentrioles in this species, Vfl1p is recruited early and at a fixed position at

the distal end of procentrioles (*Figure 3a*; *Geimer and Melkonian, 2004*; *Silflow et al., 2001*). We therefore wondered whether this mechanism might be to some extent conserved at the centrosome, which could explain the persistence of centriole rotational asymmetry despite the absence of asymmetric appendages or ciliary motility in most animal cell types. We first analyzed the timing of LRRCC1 incorporation into procentrioles. LRRCC1 was already present at an early stage of centriole assembly when the procentrioles stained with acetylated tubulin and the cartwheel component SAS-6 were only about 100 nm in length (*Figure 3b*). LRRCC1 was then detected during successive stages of procentriole elongation, always localizing asymmetrically and distally (*Figure 3c*), like in *C. reinhardtii*. We then examined LRRCC1 localization in duplicating centrosomes by generating 3D reconstructions of diplosomes (i.e., orthogonal centriole pairs) from RPE1 and HEK 293 cells processed by U-ExM (*Figure 3d*). We analyzed two parameters: the angle between LRRCC1 in the procentriole and the long axis of the parent centriole used as reference (*Figure 3d*, LRRCC1 localization in procentrioles), and the angle between procentriole position and LRRCC1 in the parent centriole (*Figure 3d*, procentriole position with respect to centriolar LRRCC1). We found that LRRCC1 localization in procentrioles was more often aligned with the long axis of the parent centriole in RPE1 cells (*Figure 3d*, top-left panel, quadrants Q1 and Q3, respectively), but less so in HEK 293 cells (top-right panel), in which the distribution was closer to a random distribution. Thus, human procentrioles do not arise in a fixed orientation, although there appears to be a bias toward alignment of LRRCC1 with the main axis of the parent centriole in RPE1 cells. Next, we analyzed the position of procentrioles with respect to centriolar LRRCC1 (bottom panels). Based on current models, procentriole assembly is expected to occur at a random position around parent centrioles in animal cells (*Takao et al., 2019*). Identification of LRRCC1 provided the first opportunity to directly test this model. In diplosomes from both RPE1 and HEK 293 cells, the position of procentrioles with respect to LRRCC1 location in the parent centriole was variable, confirming that the position at which procentrioles assemble is not strictly controlled in human cells. Interestingly, however, the procentrioles were not distributed in a completely random fashion either. Procentrioles were found in quadrant Q2 (45–135° clockwise from LRRCC1 centroid) on average four times less often than in the other quadrants, both in RPE1 and HEK 293 cells, suggesting that rotational polarity of the parent centriole somehow impacts procentriole assembly.

Overall, these results suggest that centriole rotational polarity influences centriole duplication, limiting procentriole assembly within a particular region of centriole periphery. Nevertheless, procentrioles are not formed at a strictly determined position, suggesting that the mechanisms involving the LRRCC1 ortholog Vfl1p in centriole duplication in *C. reinhardtii* are not or not completely conserved at the centrosome.

## LRRCC1 is required for primary cilium assembly and ciliary signaling

A previous report identified a homozygous mutation in a splice acceptor site of the *LRRCC1* gene in two siblings diagnosed with JBTS (*Shaheen et al., 2016*), but how disruption of LRRCC1 expression affects ciliary assembly and signaling has never been investigated. To address this, we generated RPE1 cell lines deficient in LRRCC1 using two different CRISPR/Cas9 strategies and targeting two different regions of the *LRRCC1* locus. We could not recover null clones despite repeated attempts in RPE1 – both wildtype and p53$^{-/-}$ (*Izquierdo et al., 2014*), HEK 293 and U2-OS cells, suggesting that a complete lack of LRRCC1 is possibly deleterious. Nevertheless, we obtained partially depleted mutant clones, including three RPE1 clones targeted in either exons 8–9 (clone 1.1) or exons 11–12 (clones 1.2 and 1.9). Clone 1.1 carries deletions in both copies of the *LRRCC1* gene (*Figure 4—figure supplement 1a*). However, long in-frame transcripts are expressed at reduced levels through alternative splicing (*Figure 1—figure supplement 1c*). These transcripts are expected to generate mutant protein isoforms carrying deletions in the beginning of the coiled-coil region (*Figure 4—figure supplement 1*). In contrast, only wildtype transcripts were detected in clones 1.2 and 1.9, which were present at approximately 30% of wildtype levels, as determined by quantitative RT-PCR (*Figure 1—figure supplement 1c*). We could not evaluate the overall decrease in LRRCC1 levels since the endogenous LRRCC1 protein was not detected by Western blot (*Figure 1—figure supplement 1b*). However, we confirmed the decrease in centrosomal LRRCC1 levels by immunofluorescence using the two different anti-LRRCC1 antibodies (*Figure 4a*, *Figure 1—figure supplement 1d and e*). The downregulation of LRRCC1 in CRISPR clones was overall of the same order as that achieved by RNAi, although treatment of CRISPR clones with the more efficient siRNA (si LRRCC1-1) could further

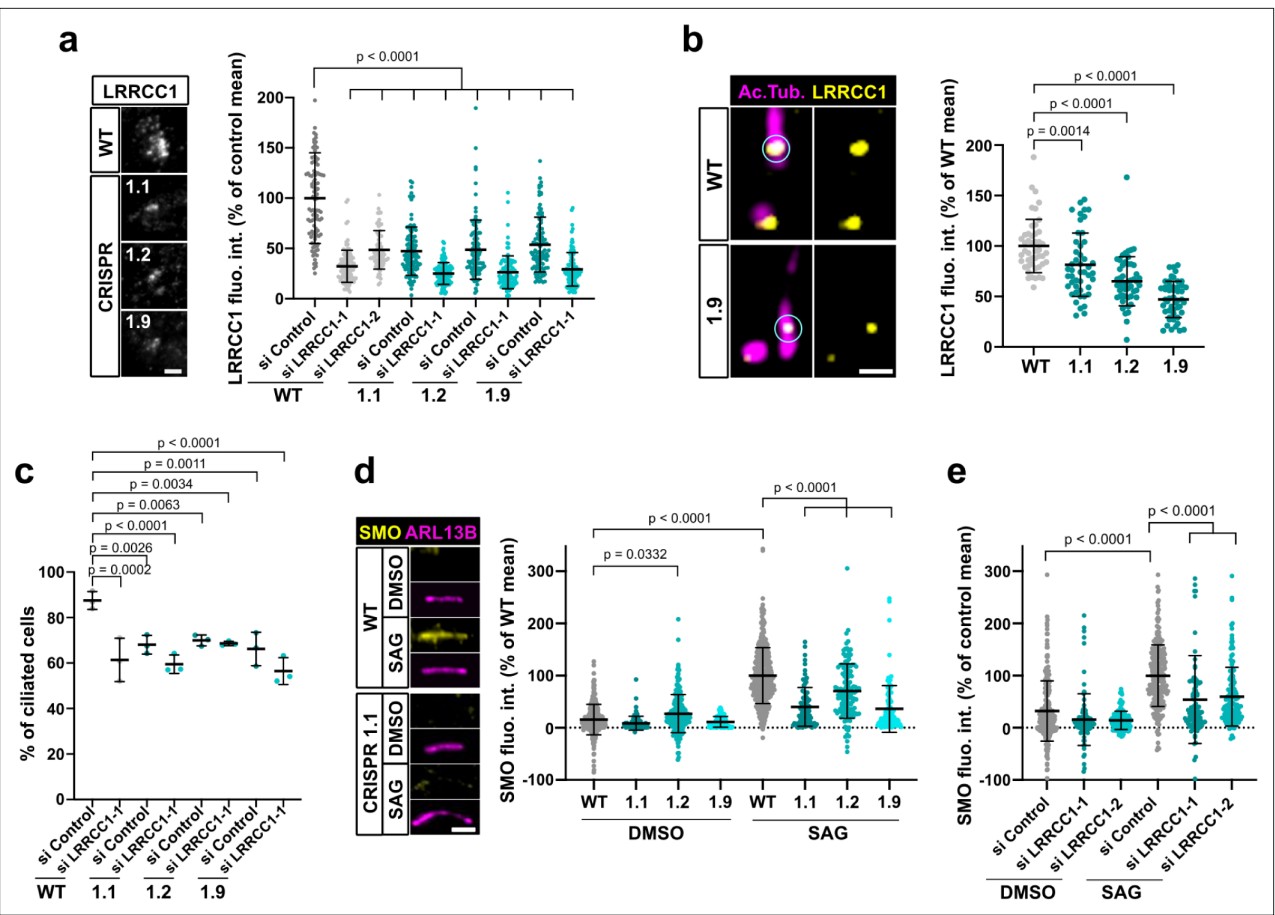

**Figure 4.** LRRCC1 is required for ciliary assembly and signaling. (**a**) Left: LRRCC1 staining (Ab2) of WT or LRRCC1-defficient RPE1 cells obtained by CRISPR/Cas9 editing (clones 1.1, 1.2, and 1.9). Bar, 2 μm. Right: quantification of fluorescence intensity in WT or CRISPR clones treated with control or LRRCC1 siRNAs. Bars, mean ± SD, three independent experiments. p-Values are provided when statistically significant from the corresponding control (one-way ANOVA). (**b**) Quantification of LRRCC1 distal pool at the mother centriole of ciliated WT or CRISPR cells. Left: Airyscan images showing the region of interest (circled). LRRCC1 (yellow), acetylated tubulin (magenta). Bar: 0.5 μm. Right: quantification of the corresponding signal. Bars, mean ± SD, ≥47 cells from two independent experiments. p-Values are provided when statistically significant from the corresponding control (one-way ANOVA). (**c**) Percentage of ciliated cells in WT or CRISPR cells treated with control or LRRCC1 siRNAs and serum-deprived during 24 hr. Bars, mean ± SD, ≥204 cells from three independent experiments for each condition. p-Values are provided when statistically significant from the corresponding control (one-way ANOVA). (**d**) Left: SMO (yellow) accumulation at primary cilia (ARL13B, magenta) following SMO-agonist (SAG)-induction of the Hedgehog pathway, in WT or CRISPR cells. Bar, 2 μm. Right: quantification of ciliary SMO expressed as a percentage of the SAG-treated WT mean. Bars, mean ± SD, three independent experiments. p-Values are provided when statistically significant from the corresponding control (one-way ANOVA). (**e**) Ciliary SMO expressed as a percentage of the SAG-induced control mean in RPE1 cells treated with control or LRRCC1 siRNAs. Bars, mean ± SD, three independent experiments. p-Values are provided when statistically significant from the corresponding control (one-way ANOVA).

The online version of this article includes the following figure supplement(s) for figure 4:

**Figure supplement 1.** Genomic characterization of the 1.1 CRISPR cell line and analysis of the corresponding transcripts.

reduce LRRCC1 levels (*Figure 4a*). Using Airyscan microscopy, we showed that LRRCC1 amounts were decreased not only at centriolar satellites, but also at the centrioles themselves in CRISPR clones (*Figure 4b*). Interestingly, the decrease in centriolar LRRCC1 was less for clone 1.1 than for the other clones, suggesting that the mutant isoforms produced in this clone have different dynamics than wild-type LRRCC1. Following induction of ciliogenesis, the proportion of ciliated cells was decreased in all three mutant clones compared to control cells (*Figure 4c*). We were unable to obtain stable RPE1 cell lines expressing tagged versions of LRRCC1, and transient overexpression of LRRCC1 in wildtype cells led to a decrease in the proportion of ciliated cells, making phenotype rescue experiments diffi-cult to interpret. However, we used RNAi as an independent method to verify the specificity of ciliary defects observed in CRISPR clones. The proportion of ciliated cells was decreased by RNAi to a similar extent than in CRISPR clones (*Figure 4c*, *Figure 1—figure supplement 1f*). RNAi treatment of CRISPR

clones did not lead to a greater decrease in ciliary frequency, suggesting that loss of LRRCC1 only partially inhibits ciliogenesis (*Figure 4c*). Sensory ciliopathies like JBTS result to a large extent from defective Hedgehog signaling (*Romani et al., 2013*). We determined the effect of LRRCC1-depletion on Hedgehog signaling by measuring the ciliary accumulation of the activator SMOOTHENED (SMO) upon induction of the pathway (*Rohatgi et al., 2007*). Depletion of LRRCC1 by either CRISPR editing or RNAi led to a drastic decrease in SMO accumulation at the primary cilium following induction of the Hedgehog pathway by SMO-agonist (SAG) (*Figure 4d and e*), and reduced expression of the target gene *PTCH1* (*Figure 1—figure supplement 1i*; *Goodrich et al., 1996*). Taken together, our results demonstrate that LRRCC1 is required for proper ciliary assembly and signaling in human cells, further establishing its implication in JBTS.

## Depletion of LRRCC1 induces defects in centriole structure

Mutations in distal centriole components can alter centriole length regulation or the assembly of DAs, which both result in defective ciliogenesis (*Reiter and Leroux, 2017*; *Sharma et al., 2021*). We used U-ExM to search for possible defects in centriole structure in LRRCC1-depleted RPE1 cells. We measured centriole length in CRISPR clone 1.9, which has the lowest levels of centriolar LRRCC1 (*Figure 4b*), and in clone 1.1, which expresses mutant isoforms of LRRCC1. Centrioles were co-stained with anti-acetylated tubulin and an antibody against the DA component CEP164 to differentiate mother and daughter centrioles. We observed an increase in centriole length in clone 1.9 (*Figure 5a*) compared to control cells (483 ± 53 nm for mother and 372 ± 55 nm for daughter centrioles in clone 1.9; 427 ± 56 nm for mother and 320 ± 46 nm for daughter centrioles in control cells; mean ± SD). Although on a limited sample size, we also observed abnormally long centrioles by transmission electron microscopy in this clone (494 ± 73 nm in clone 1.9, N = 9; 429 ± 52 nm in control cells, N = 3; mean ± SD) (*Figure 5c*). The increase in centriole length was not due to mitotic delay as previously observed (*Kong et al., 2020*) since the duration of mitosis in clone 1.9 was similar as in control cells (*Figure 1—figure supplement 1k*). In addition, although centriole length was not modified in clone 1.1, further reduction of LRRCC1 levels by RNAi resulted in a significant increase in centriole length compared to control cells (*Figure 5b*). Next, we analyzed DA organization by labeling CEP164, which localizes to the outer part of DAs (*Figure 5d*; *Yang et al., 2018*). In RPE1 control cells, 80% ± 14% of mother centriole had nine properly shaped DAs, but this proportion fell to 57% ± 16% and 44% ± 17% (mean ± SD) in clones 1.1 and 1.9, respectively (*Figure 5e*). Mutant clones exhibited an increased proportion of centrioles with one or more abnormally shaped DAs (29% ± 17% and 42% ± 18% in clones 1.1 and 1.9, respectively, compared to 11% ± 11% in control cells; mean ± SD). We obtained similar results in a HEK 293 CRISPR clone expressing half the control levels of LRRCC1 (*Figure 5f*, *Figure 1—figure supplement 1g*). LRRCC1 depletion did not affect overall CEP164 levels at mother centrioles in the CRISPR clones (*Figure 5—figure supplement 1a and d*), consistent with the relatively mild defect in DA morphology observed by U-ExM. We also analyzed the distribution of CEP83, a DA component that localizes closer to the centriole wall (*Yang et al., 2018*). The proportion of centrioles with abnormal CEP83 labeling was not significantly different between control cells and CRISPR clones. However, this proportion became significantly lower than in the control after treating CRISPR clones with RNAi (41% ± 18% and 48% ± 4% in clones 1.1 and 1.9 treated with RNAi, respectively, compared to 77% ± 9% in control cells; mean ± SD; *Figure 5g and h*). Beyond these anomalies in centriolar structure, LRRCC1-depleted cells showed no defect in centriole number, supporting that centriole assembly is not affected by LRRCC1 downregulation (*Figure 1—figure supplement 1j*).

Together, these results show that downregulation of LRRCC1 affects the formation of centriole distal structures, leading to centriole over-elongation and abnormal DA morphology.

## LRRCC1 and C2CD3 delineate a rotationally asymmetric structure in human centrioles

We next wanted to determine whether LRRCC1 cooperates with other distal centriole components. Proteins shown to be recruited early at procentriole distal end include CEP290 (*Kim et al., 2008*), OFD1 (*Singla et al., 2010*), and C2CD3 (*Thauvin-Robinet et al., 2014*). Of particular interest, OFD1 and C2CD3 are required for DA assembly and centriole length control, and mutations in these proteins have been implicated in sensory ciliopathies (*Singla et al., 2010*; *Thauvin-Robinet et al., 2014*; *Tsai et al., 2019*; *Wang et al., 2018*). We first determined whether depletion of LRRCC1 either by CRISPR

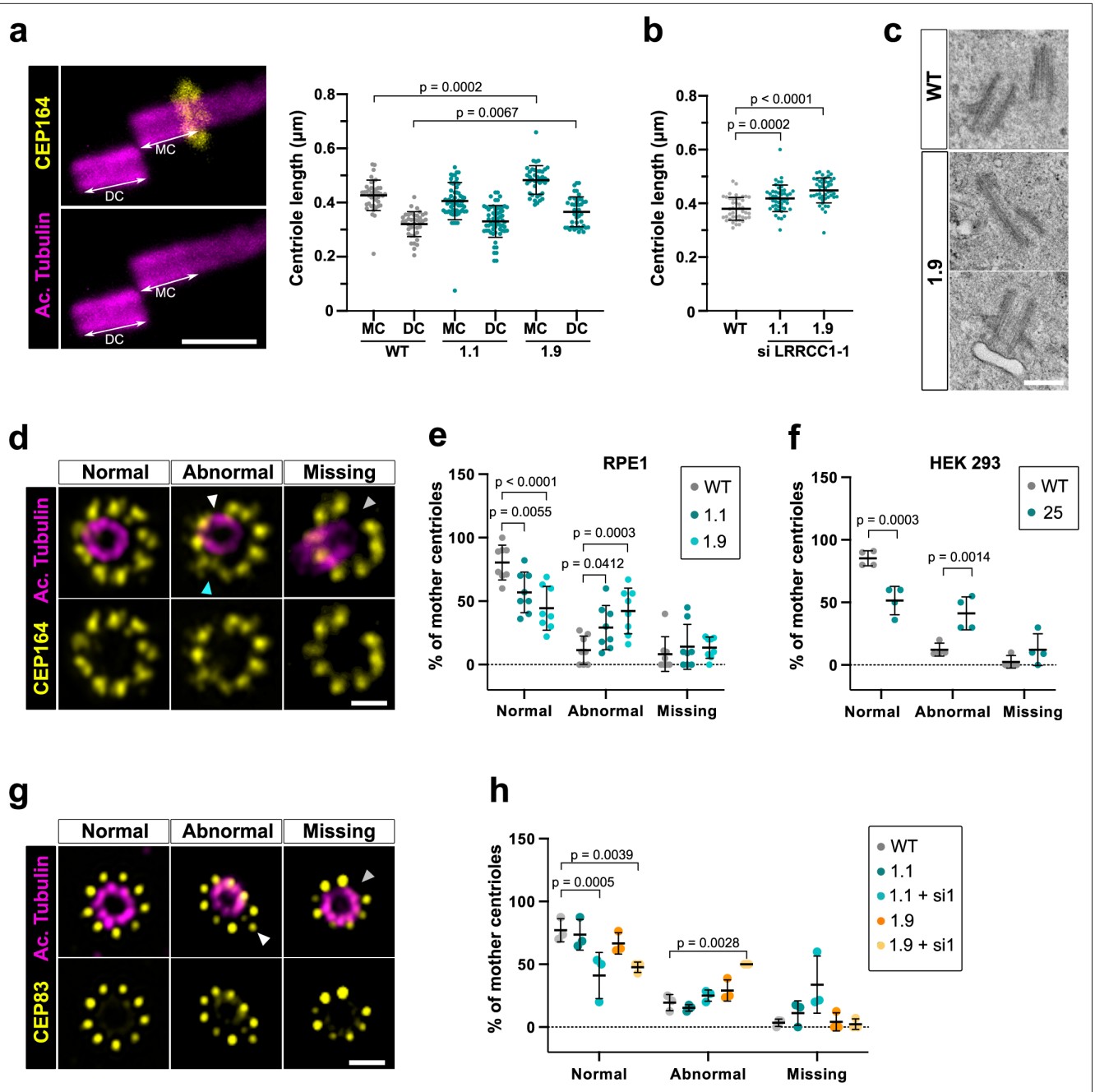

**Figure 5.** Depleting LRRCC1 induces defects in centriole structure. (**a**) Centriole length in mother (MC) and daughter (DC) centrioles analyzed by ultrastructure expansion microscopy (U-ExM) in WT or LRRCC1-deficient clones (1.1 and 1.9). Left: centrioles were stained for acetylated tubulin (magenta) and CEP164 (yellow) to measure centriole length (arrows). Bar, 0.5 μm. Right: quantification. Bars, mean ± SD, ≥38 centrioles from three independent experiments. p-Values are provided when statistically significant from the corresponding control (one-way ANOVA). (**b**) Centriole length in control cells or CRISPR cells treated with LRRCC1 siRNA-1 and stained for acetylated tubulin and CEP83. Bars, mean ± SD, ≥43 centrioles from three independent experiments. p-Values are provided when statistically significant from the corresponding control (one-way ANOVA). (**c**) Transmission electron microscopy view of centrioles in WT and CRISPR (clone 1.9) RPE1 cells. Note that the 1.9 centrioles are from the same cell. N = 9 centrioles from eight different cells for clone 1.9, 3 centrioles from two different cells for WT. Bar, 0.5 μm. (**d**) Examples of normal distal appendages (DAs), DAs with abnormal morphology (white arrowhead: abnormal spacing between consecutive DAs; cyan arrowhead: abnormal DA shape) or missing DAs (gray arrowhead) in RPE1 cells stained with CEP164 (yellow) and analyzed by U-ExM. Images are maximum intensity projections of individual z-sections encompassing the CEP164 signal. Note that an apparent shift between channels occurs when centrioles are slightly angled with respect to the imaging axis. Bar, 1 μm. (**e**) Percentages of centrioles presenting anomalies in CEP164 staining in WT or CRISPR RPE1 cells. ≥87 centrioles from eight independent experiments for each condition. p-Values are provided when statistically significant from the corresponding control (two-way ANOVA).

*Figure 5 continued on next page*

*Figure 5 continued*

(**f**) Percentages of centrioles presenting anomalies in CEP164 staining in WT or CRISPR HEK 293 (clone 25) cells. ≥40 centrioles from four independent experiments for each condition. p-Values are provided when statistically significant from the corresponding control (two-way ANOVA). (**g**) Examples of normal DAs, DAs with abnormal morphology (white arrowhead) or missing DAs (gray arrowhead) in RPE1 cells stained with CEP83 (yellow) and analyzed by U-ExM. Images are maximum intensity projections of individual z-sections encompassing the CEP83 signal. Note that apparent shift between channels and decreased circularity occurs when centrioles are slightly angled with respect to the imaging axis. Bar, 1 μm. (**h**) Percentages of centrioles presenting anomalies in CEP83 staining in WT RPE1 cells and CRISPR clones with or without RNAi treatment. ≥56 centrioles from three independent experiments for each condition. p-Values are provided when statistically significant from the corresponding control (two-way ANOVA).

The online version of this article includes the following figure supplement(s) for figure 5:

**Figure supplement 1.** Quantification of distal appendage (DA) or distal centriole components in LRRCC1-deficient cells.

editing or by RNAi led to modifications in the recruitment of these proteins within centrioles. We found no major differences in the centrosomal levels of OFD1 and CEP290 compared to control cells (*Figure 5—figure supplement 1b, c, e, and f*). In contrast, C2CD3 levels were moderately increased in cells depleted from LRRCC1 either by CRISPR editing (clones 1.1 and 1.9) or by RNAi (*Figure 6a and b*). We thus analyzed C2CD3 further by U-ExM. As described previously, C2CD3 localized principally at the distal extremity of centrioles (*Figure 6c*; *Tsai et al., 2019*; *Yang et al., 2018*). Strikingly, the C2CD3 labeling was also asymmetric, often adopting a C-shape (*Figure 6d*). After correcting the vertical alignment of centrioles as previously, we generated an average 3D reconstruction of the C2CD3 pattern. To do this, we used one end of the C as a reference point in the xy-plane to superimpose individual centriole views. The resulting image supported that the C2CD3 labeling forms a C-shaped pattern positioned symmetrically in the centriole lumen (*Figure 6e*). To determine whether the C2CD3 localization pattern is affected by LRRCC1-depletion, we next analyzed C2CD3 in LRRCC1 CRISPR clones 1.1 and 1.9. The C2CD3 pattern was more variable than in control RPE1 cells, and often appeared abnormal in shape, position, or both (*Figure 6f*). Indeed, averaging the signal from multiple LRRCC1-depleted centrioles produced aberrant patterns, most strikingly for clone 1.9 (*Figure 6g*). Furthermore, the phenotype of clone 1.1 was enhanced by further reducing LRRCC1 levels using RNAi (*Figure 6g*). Thus, LRRCC1 is required for the proper assembly of the C2CD3-containing distal structure.

To determine whether LRRCC1 and C2CD3 might belong to a common structure, we next examined their respective positions within the centriole. We co-stained centrioles with our anti-LRRCC1 antibody and a second anti-C2CD3 antibody produced in sheep (*Table 1*). We confirmed that LRRCC1 and C2CD3 are present in the same distal region of the centriole (*Figure 7a*). In transverse views, the two proteins were usually not perfectly colocalized but found in close vicinity of one another near the microtubule wall. However, C2CD3 distal staining was consistently fainter than with the previous antibody, and we either could not observe a full C-shaped pattern or we could not image it due to fluorescence bleaching. Neither anti-C2CD3 antibodies worked in mouse, so we were not able to compare C2cd3 and Lrrcc1 localization in MCCs. Nevertheless, the results obtained by individually labeling LRRCC1 and C2CD3 at the centrosome (*Figures 1f and 6e*) together with the co-localization data (*Figure 7a*) are consistent with the hypothesis that LRRCC1 is located along the C2CD3-containing, C-shaped structure (*Figure 7b*). C2CD3 was not co-immunoprecipitated with a GFP-LRRCC1 fusion protein, however, suggesting that LRRCC1 and C2CD3 do not directly interact (*Figure 7—figure supplement 1*).

Taken together, our results support that C2CD3 localizes asymmetrically in the distal lumen of human centrioles, a pattern that depends in part on LRRCC1.

## Discussion

Here, we show that centrioles within the human centrosome are rotationally asymmetric despite the apparent ninefold symmetry of their ultrastructure. This asymmetry is manifested by a specific enrichment in LRRCC1 near two consecutive triplets, and the C-shaped pattern of C2CD3. Depletion of LRRCC1 perturbed the recruitment of C2CD3 and induced defects in centriole structure, ciliogenesis, and ciliary signaling, supporting that LRRCC1 contributes to organizing the distal centriole region together with C2CD3. LRRCC1 localizes like its *C. reinhardtii* ortholog Vfl1p, and C2CD3 delineates a filamentous structure reminiscent of the acorn first described in *C. reinhardtii* (*Geimer and Melkonian,*

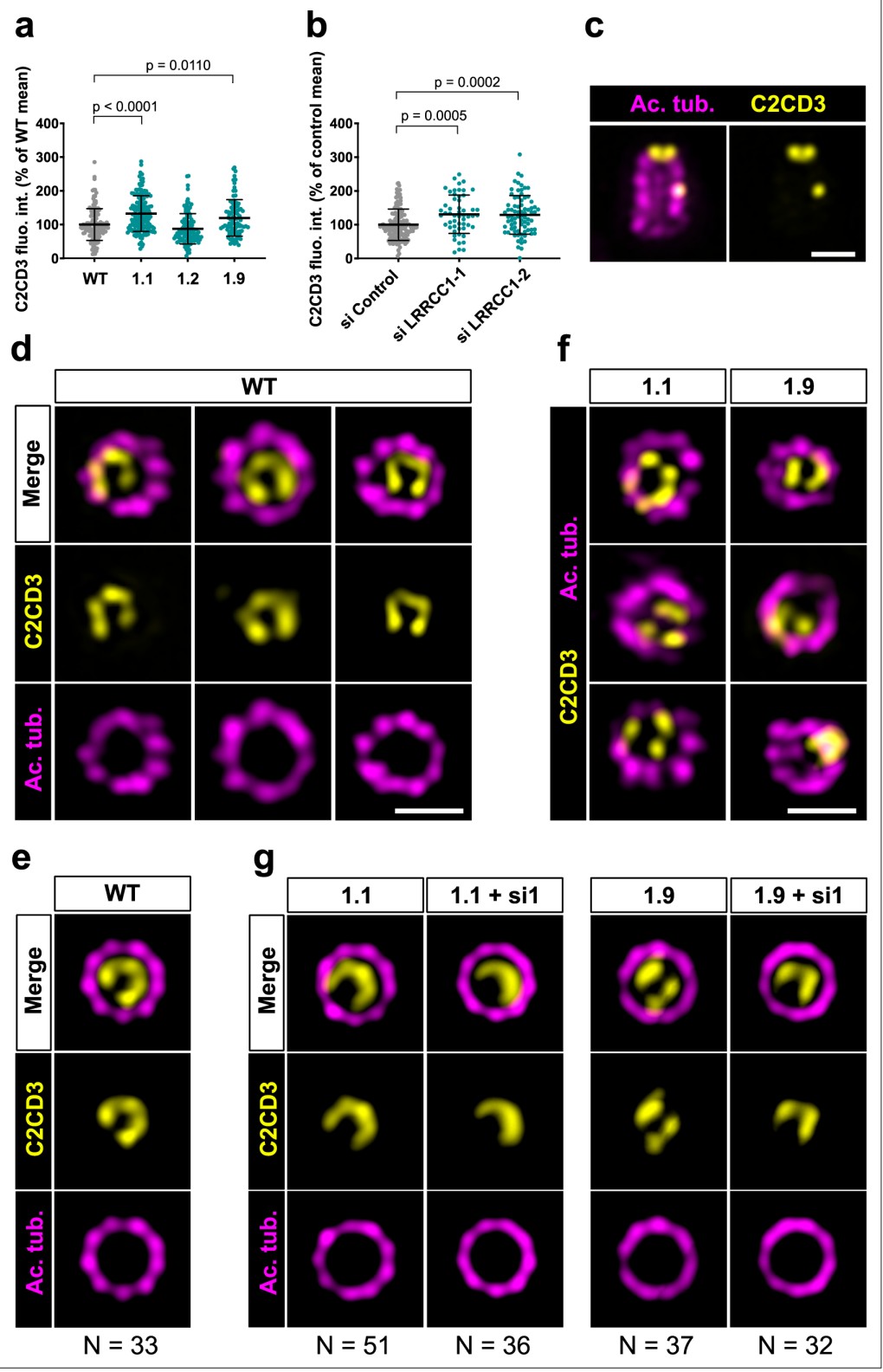

**Figure 6.** C2CD3 localizes asymmetrically at the distal end of centrioles and is affected by LRRCC1 depletion. (**a**) C2CD3 levels at the centrosome of WT or CRISPR RPE1 cells. Bars, mean ± SD, three independent experiments. p-Values are provided when statistically significant from the corresponding control (one-way ANOVA). (**b**) C2CD3 levels at the centrosome in RPE1 cells treated with control or LRRCC1 siRNAs. Bars, mean ± SD, three independent experiments. p-Values are provided when statistically significant from the corresponding control (one-way

*Figure 6 continued on next page*

*Figure 6 continued*

ANOVA). (**c**) Longitudinal view of a centriole analyzed by ultrastructure expansion microscopy (U-ExM) and stained for C2CD3 (yellow) and acetylated tubulin (magenta). Bar, 0.2 μm. (**d**) Centrioles from WT RPE1 cells as viewed from the distal end. C2CD3 (yellow), acetylated tubulin (magenta). Images are maximum intensity projections of individual z-sections encompassing the C2CD3 signal. Note that an apparent shift between channels occurs when centrioles are slightly angled with respect to the imaging axis. Bar, 0.2 μm. (**e**) Average C2CD3 images obtained from 33 individual centrioles from WT RPE1 cells viewed from the distal end, in transverse views. One end of the C-pattern was used as a reference point to align individual centrioles. (**f**) Centrioles from untreated CRISPR cells or CRISPR cells treated with LRRCC1 RNAi in transverse section as viewed from the distal end. C2CD3 (yellow), acetylated tubulin (magenta). Images are maximum intensity projections of individual z-sections encompassing the C2CD3 signal. Note that an apparent shift between channels occurs when centrioles are slightly angled with respect to the imaging axis. Bar, 0.2 μm. (**g**) Average C2CD3 images obtained from untreated or RNAi-treated CRISPR cells viewed from the distal end, in transverse views. The number or individual centrioles used for generating each average is indicated.

2005; *Geimer and Melkonian, 2004*) and later found in a wide variety of eukaryotic species (*Cavalier-Smith, 2021*; *Vaughan and Gull, 2015*). Collectively, our results support that rotational asymmetry is a conserved property of centrioles linked to ciliary assembly and signaling in humans.

## LRRCC1 and C2CD3 belong to a conserved rotationally asymmetric structure

Our work identifies two proteins located asymmetrically in the distal centriole lumen of the human centrosome, each with a specific pattern. LRRCC1 localizes principally near two consecutive triplets, with the first triplet counterclockwise labeled approximately 50% more than the next one. This pattern is highly reminiscent of the LRRCC1 ortholog Vfl1p, which localizes predominantly to the triplet facing the second centriole (referred to as triplet 1), and to a lesser extent to its immediate neighbor counterclockwise (triplet 2; *Figure 7b*; *Silflow et al., 2001*). In *C. reinhardtii*, triplets 1 and 2 are positioned directly opposite to the direction of flagellar beat, which is directed towards triplet 6 (*Figure 7b*; *Lin et al., 2012*). In mouse MCCs, Lrrcc1 is associated to triplets located not exactly opposite to the basal foot but with a clockwise shift of at least 20° from the basal foot axis. However, the beating direction was shown to be also shifted approximately 20° clockwise relative to the position of the basal foot in bovine tracheal MCCs (*Schneiter et al., 2021*; *Figure 2d*). The position of Lrrcc1/Vfl1p-labeled triplets with respect to ciliary beat direction might thus be similar in *C. reinhardtii* and in animal MCCs. Overall, the specific localization pattern of Vfl1p-related proteins at the distal end of centrioles, and their requirement for centriole positioning and ciliary beat orientation when motile cilia are present, appears to be conserved between flagellates and animals.

The second protein conferring rotational asymmetry to human centrioles, C2CD3, delineates a C-shape in the distal lumen. Strikingly, this staining is reminiscent of a filament observed by electron microscopy, which is said to form an 'incomplete circle' in the distal lumen of human centrioles (*Vorobjev and Chentsov, 1980*). Several lines of evidence favor the hypothesis that the C2CD3-containing structure is homologous to the acorn, a conserved filamentous structure that in *C. reinhardtii* connects five consecutive triplets along the centriole wall and across the lumen (*Figure 7b*; *Cavalier-Smith, 2021*; *Geimer and Melkonian, 2004*; *Vaughan and Gull, 2015*). First, the C2CD3 labeling is consistent with a circular filament. Second, C2CD3 is partially co-localized with LRRCC1 near the microtubule wall. Lastly, C2CD3 orthologs are found in a variety of flagellated unicellular eukaryotes, including the green algae *Micromonas pusilla* (*Zhang and Aravind, 2012*) and *Chlamydomonas eustigma* (Uniprot_A0A250XH15), suggesting an ancestral association to centrioles and cilia. The partial co-localization of Vfl1p and the acorn in *C. reinhardtii*, and the observation that both are already present at the distal end of procentrioles, led to propose that Vfl1p might also be a component of the acorn (*Geimer and Melkonian, 2004*). Consistent with this idea, both LRRCC1 and C2CD3 are recruited early to the distal end of human procentrioles, and LRRCC1 is required for proper assembly of the C2CD3-containing structure. C2CD3 recruitment at the centrioles also depends on the proteins CEP120 and Talpid3 (*Tsai et al., 2019*). Future work will help deciphering the relationships between these different proteins and characterize in more detail the architecture of the rotationally asymmetric structure at the distal end of mammalian centrioles.

**Table 1.** Antibodies used in this study.

| Antibody | Dilution IF | Dilution U-ExM | Dilution WB | RRID identifier | Source | Reference |
|---|---|---|---|---|---|---|
| **Primary antibodies** | | | | | | |
| Goat anti-ARL13B | 1:100 | / | / | RRID:AB_2058502 | Santa Cruz Biotechnology | sc-102318 |
| Guinea pig anti-alpha tubulin AA344 monobody | / | 1:500 | / | | Geneva Antibody Facility | scFv-S11B |
| Guinea pig anti-beta tubulin AA345 monobody | / | 1:500 | / | | Geneva Antibody Facility | scFv-F2C |
| Mouse anti-acetylated tubulin (6-11B-1) | 1:1000 | 1:500 | / | RRID:AB_628409 | Santa Cruz Biotechnology | sc-23950 |
| Mouse anti-Centrin (20H5) | 1:500 | / | / | RRID:AB_10563501 | Sigma-Aldrich | 04–1624 |
| Mouse anti-CEP290 (B-7) | 1:500 | / | / | RRID:AB_2890036 | Santa Cruz Biotechnology | sc-390462 |
| Mouse anti-gamma tubulin (GTU88) | 1:2000 | 1:200 | / | RRID:AB_532292 | Sigma-Aldrich | T5326 |
| Mouse anti-SAS-6 | / | 1:100 | / | RRID:AB_1128357 | Santa Cruz Biotechnology | sc-81431 |
| Mouse anti-Smoothened | 1:200 | / | / | RRID:AB_1270802 | Abcam | ab72130 |
| Rabbit anti-ARL13B | 1:500 | / | / | RRID:AB_2060867 | Proteintech | 17711–1-AP |
| Rabbit anti-C2CD3 | 1:500 | 1:500 | / | RRID:AB_10669542 | Sigma-Aldrich | HPA038552 |
| Rabbit anti-C2CD3 | / | / | 1:1000 | RRID:AB_2718714 | Thermo Fisher Scientific | PA5-72860 |
| Rabbit anti-CEP83 | / | 1:500 | / | RRID:AB_10674547 | Sigma-Aldrich | HPA038161 |
| Rabbit anti-CEP164 | 1:500 | 1:300 | / | RRID:AB_2651175 | Proteintech | 22227–1-AP |
| Rabbit anti-GFP | / | / | 1:1000 | RRID:AB_591816 | MBL International | 598 |
| Rabbit anti-HA | / | / | 1:1000 | RRID:AB_631618 | Santa Cruz Biotechnology | sc-805 |
| Rabbit anti-hPOC5 | 1:500 | / | / | | *Azimzadeh et al., 2009* | |
| Rabbit anti-KI67 | 1:1000 | / | / | RRID:AB_443209 | Abcam | ab15580 |
| Rabbit anti-LRRCC1 Ab1 | 1:500 | 1:200 | / | | This study | |
| Rabbit anti-LRRCC1 Ab2 | 1:500 | 1:300 | 1:1000 | | This study | |
| Rabbit anti-OFD1 | 1:500 | / | / | RRID:AB_2890033. | Sigma-Aldrich | ABC961 |
| Sheep anti-C2CD3 | 1:200 | 1:100 | / | RRID:AB_10997138 | R&D Systems | AF7348 |
| **Secondary antibodies** | | | | | | |
| Donkey anti-goat IgG H&L (Alexa Fluor 488) | 1:500 | 1:500 | / | RRID:AB_2687506 | Abcam | ab150129 |
| Donkey anti-goat IgG H&L (Alexa Fluor 568) | 1:500 | 1:500 | / | RRID:AB_2636995 | Abcam | ab175474 |
| Donkey anti-goat IgG H&L (Alexa Fluor 647) | 1:500 | 1:100 | / | RRID:AB_2732857 | Abcam | ab150131 |
| Donkey anti-mouse IgG H&L (Alexa Fluor 488) | 1:500 | 1:500 | / | RRID:AB_2732856 | Abcam | ab150105 |
| Donkey Anti-Mouse IgG H&L (Alexa Fluor 568) | 1:500 | 1:500 | / | RRID:AB_2636996 | Abcam | ab175472 |
| Donkey anti-mouse IgG H&L (Alexa Fluor 647) | 1:500 | 1:100 | / | RRID:AB_2890037 | Abcam | ab150107 |
| Donkey anti-rabbit IgG H&L (Alexa Fluor 488) | 1:500 | 1:500 | / | RRID:AB_2636877 | Abcam | ab150073 |
| Donkey anti-rabbit IgG H&L (Alexa Fluor 647) | 1:500 | 1:500 | / | RRID:AB_2752244 | Abcam | ab150075 |
| Donkey anti-sheep IgG H&L (Alexa Fluor 647) | / | 1:100 | / | RRID:AB_2884038 | Abcam | ab150179 |
| Goat anti-guinea pig IgG (H+L) (Alexa Fluor 568) | / | 1:100 | / | RRID:AB_141954 | Thermo Fisher Scientific | A-11075 |
| Goat anti-rabbit IgG (H+L) horseradish peroxidase conjugate | / | / | 1:1000 | RRID:AB_2536530 | Thermo Fisher Scientific | G-21234 |

*Table 1 continued on next page*

*Table 1 continued*

| Antibody | Dilution IF | Dilution U-ExM | Dilution WB | RRID identifier | Source | Reference |
|---|---|---|---|---|---|---|

IF: immunofluorescence; U-ExM: ultrastructure expansion microscopy WB: Western blot.

## Rotationally asymmetric centriole components are required for ciliogenesis

Our results uncover a link between centriole rotational asymmetry and primary ciliogenesis in human cells. Mutations in C2CD3 have been involved in several sensory ciliopathies, including JBTS (*Boczek et al., 2018*; *Cortés et al., 2016*; *Ooi, 2015*; *Thauvin-Robinet et al., 2014*). The associated ciliary defects are likely caused by anomalies in the structure of centrioles since depleting C2CD3 inhibits centriole elongation and DA assembly, whereas C2CD3 overexpression leads to centriole hyper-elongation (*Thauvin-Robinet et al., 2014*; *Wang et al., 2018*; *Ye et al., 2014*). We observed similar defects in LRRCC1-depleted cells, but of comparatively lesser extent. DA morphology was altered and centriole length was slightly increased in cells depleted from LRRCC1. The fact that LRRCC1 deple-tion has a more limited impact on centriole assembly than perturbation of C2CD3 levels suggests that LRRCC1 might not be directly involved in centriole length control or DA formation, however. The defects observed in LRRCC1-depleted cells could instead result indirectly from the abnormal localization of C2CD3. Besides the defects in centriole structure, it is plausible that LRRCC1 depletion also perturbs the organization of the ciliary gate as LRRCC1-depleted cells exhibited a drastic reduc-tion in Hedgehog signaling. Loss of ciliary gate integrity interferes with the accumulation of SMO in the cilium upon activation of the Hedgehog pathway and is a frequent consequence of ciliopathic mutations (*Garcia-Gonzalo and Reiter, 2017*). The ciliary gate consists of the (Transition zone) TZ and the DA region, which both contribute to regulating protein trafficking in and out of the cilium (*Garcia-Gonzalo and Reiter, 2017*; *Nachury, 2018*). The anomalies in DA morphology observed in LRRCC1-depleted cells could disrupt the organization of the so-called DA matrix (*Yang et al., 2018*), thus preventing SMO accumulation in the cilium. Another, nonmutually exclusive possibility is that the architecture of the TZ, which forms directly in contact with the distal end of the centriole, is altered by LRRCC1 depletion. In either case, our observations in RPE1 cells are consistent with the JBTS diagnosis in two siblings carrying a mutation in the *LRRCC1* gene (*Shaheen et al., 2016*), further establishing that *LRRCC1* is a novel ciliopathy gene. Besides signaling, ciliary gate integrity is required for axoneme extension, and indeed, LRRCC1-depleted cells formed cilia at lower frequency than control cells – a defect that might also result from perturbed DA architecture. In the *vfl1* mutant of *C. reinhardtii*, both unanchored centrioles and centriole docked at the plasma membrane but lacking a flagellum were observed (*Adams et al., 1985*). This supports that LRRCC1/Vfl1p requirement for prop-erly assembling the ciliary gate is a conserved functional aspect of this family of proteins (*Figure 7c*).

Why is there a rotationally asymmetric structure at the base of primary cilia, and how does this structure form and contribute to the assembly of the DAs and the cilium remain open questions. In *C. reinhardtii* and in MCCs, LRRCC1 function is linked to the assembly of asymmetric appendages, which must be correctly positioned in relation to ciliary beat direction (*Figure 7c*). An asymmetric structure present early during centriole assembly and ultimately located near the cilium appears well suited for this task. The conservation of such a structure at the base of the primary cilium could perhaps indicate that primary cilia also possess rotationally asymmetric features, which would open interesting new perspectives on ciliary roles in heath and disease.

## Other roles for centriole rotational asymmetry in human cells

Our finding that procentrioles do not form completely at random with respect to LRRCC1 location in the parent centriole suggests that centriole rotational polarity can influence centriole duplication in human cells. In *C. reinhartdtii*, procentrioles are formed at fixed positions with respect to the parent centrioles, to which they are bound by a complex array of fibrous and microtubular roots (*Figure 7c*; *Geimer and Melkonian, 2004*; *Yubuki and Leander, 2013*). The process is likely different at the centrosome since the roots typical of flagellates are not conserved in animal cells (*Azimzadeh, 2021*; *Yubuki and Leander, 2013*). In mammalian cells, procentrioles form near the wall of the parent centriole following the recruitment of early centriole proteins directly to the PCM components CEP152

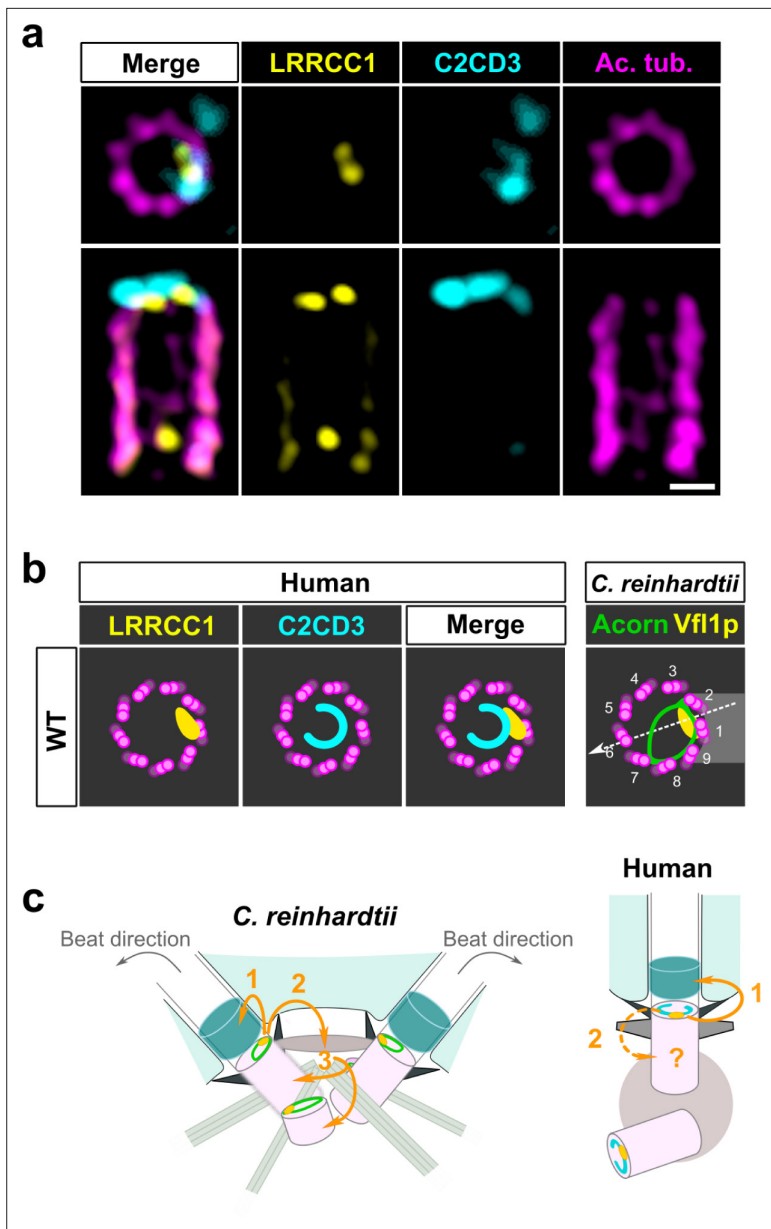

**Figure 7.** C2CD3 and LRRCC1 partially colocalize at the distal end of centrioles. (**a**) RPE1 centrioles processed for ultrastructure expansion microscopy (U-ExM) and stained for LRRCC1 (Ab2, yellow), C2CD3 (cyan), and acetylated tubulin (magenta). Bar, 0.1 µm. (**b**) Model showing the possible location of LRRCC1 and C2CD3 relative to each other within human centrioles. Right panel: diagram showing the respective positions of the acorn (*Geimer and Melkonian, 2004*) and Vfl1p (*Silflow et al., 2001*) in *C. reinhardtii*. The direction of the flagellar beat is indicated by a dotted arrow, and the distal striated fiber is in gray. (**c**) Evolution of the roles played by Vfl1p/LRRCC1 proteins and associated rotationally asymmetric centriolar substructures. In *C. reinhardtii*, Vfl1p is required for proper ciliary assembly (1), as well as for the formation of fibers and microtubular roots (2) that control the position of centrioles and procentrioles (3), and overall cellular organization (*Adams et al., 1985*; *Silflow et al., 2001*). In human cells, LRRCC1 and C2CD3 are required for primary cilium assembly (1) – this study and *Thauvin-Robinet et al., 2014*; *Ye et al., 2014* – and a role in asymmetric anchoring of cytoskeletal elements to the centriole may also be conserved (2), which could indirectly affect the determination of procentriole assembly site.

The online version of this article includes the following source data and figure supplement(s) for figure 7:

**Figure supplement 1.** LRRCC1 does not interact directly with C2CD3.

**Figure supplement 1—source data 1.** Western blot analysis of LRRCC1/C2CD3 co-immunoprecipitation assay.

and CEP192 (*Yamamoto and Kitagawa, 2021*). It is nonetheless conceivable that an asymmetry in triplet composition could result in local changes in PCM composition, which in turn could negatively impact PLK4 activation in this region. For instance, our analyses in planarian MCCs led us to postulate that linkers might be tethered to one side of the centrioles in a VFL1-dependent manner and independently of centriole appendages (*Basquin et al., 2019*). Future work will allow deciphering how centriole rotational asymmetry influences centriole duplication, and whether it affects other aspects of centriole positioning and cellular organization.

## Materials and methods

### Cell culture

RPE1 cells (hTERT-RPE1, RRID:CVCL_4388; American Type Culture Collection, authenticated by STP profiling, no mycoplasma detected) were cultured in DMEM/F-12 medium (Thermo Fisher Scientific) supplemented with 10% fetal calf serum (Thermo Fisher Scientific), 100 U/mL penicillin and 100 µg/mL streptomycin (Thermo Fisher Scientific). Ciliogenesis was induced by culturing RPE1 cells in medium without serum during 48 hr. HEK 293 cells (kind gift from F. Causeret, Institut Imagine, Paris) were cultured in DMEM medium (Thermo Fisher Scientific) supplemented with 10% fetal calf serum and antibiotics as previously. All cells were kept at 37°C in the presence of 5% $CO_2$.

### Mouse ependymal cells and tracheal tissue

All experiments were performed in accordance with the French Agricultural Ministry and European guidelines for the care and use of laboratory animals. In vitro differentiated ependymal cells were a kind gift from A.R. Boudjema and A. Meunier (IBENS, Paris). They were prepared as described previously (*Delgehyr et al., 2015*; *Mercey et al., 2019*) from Cen2GFP mice (CB6-Tg(CAG-EGFP/CETN2)3-4Jgg/J, The Jackson Laboratory). The fragment of trachea was obtained from a wildtype mouse of the Swiss background (kindly provided by I. Le Parco, Institut Jacques Monod).

### CRISPR/Cas9 editing

LRRCC1 mutant clones were obtained by two different CRISPR/Cas9 strategies. First, RPE1 cells were co-transfected with plasmid px154-1 (U6p-gRNA#1_U6p-gRNA#2_CMVpnCas9-EGFP_SV40p-PuroR-pA with gRNA#1: 5′-AGA ATT CTA CCC TAC CTG-3′ and gRNA#2: 5′-TAA GGT AGT GCT TCC TAC-3′) targeting the *LRRCC1* locus in exon 8, and px155-24 (U6p-gRNA#3_U6p-gRNA#4_CMVpnCas9-mCherry_SV40p-PuroR-pA; gRNA#1: 5′-ATC TAC TCG GAA AGC TGA-3′ and 5′-GCT TGA GGG CTC AAA TAC-3′) targeting exon 9. Both constructs express the nickase mutant of Cas9 fused to either EGFP or mCherry. Two days after transfection, EGFP- and mCherry-positive cells were sorted by flow cytometry and grown at low concentration. Individual clones were picked after 2 weeks and analyzed by PCR to detect short insertions/deletions. A single clone was obtained (clone 1.1), which was further characterized by genomic sequencing. Both alleles of *LRRCC1* contained deletions (~0.6 kb deletion of exon 9 and an ~1.5 kb deletion of exon 8; *Figure 4—figure supplement 1a*), leading to frameshifts. In a second approach, cells were co-transfected using a mix of three CRISPR/Cas9 Knockout Plasmids (sc-413781; Santa Cruz Biotechnology) targeting exons 11 (5′-CTT GTT CTC TTT CTC GAT GA-3′ and 5′-ACT TCT TGC ATT GAA AGA AC-3′) or 12 (5′-CGT GTT AAG CCA GCA GTA TA-3′) of *LRRCC1*, together with the corresponding homology-directed repair plasmids carrying a puromycin-resistance cassette (sc-413781-HDR; Santa Cruz Biotechnology), following the recommendations of the manufacturer. Mutant clones were selected by addition of 2 µg/mL puromycin in the culture medium and further screened by immunofluorescence, allowing to identify two clones with decreased LRRCC1 levels (clones 1.2 and 1.9). Genomic insertion of the HDR cassette could not be detected in these clones by PCR, and no sequence anomalies were identified in PCR fragments corresponding to exons 10–13. This suggests that one copy of the *LRRCC1* gene is intact, while the second copy may have undergone more extensive modifications via large deletions/insertions. For sequencing of LRRCC1 transcripts, total RNA extracts were obtained using the Nucleosin RNA kit (Macherey-Nagel) and cDNAs were synthesized using SuperScript III reverse transcriptase (Thermo Fisher Scientific). PCR primers specific to exons 4 and 8, 4 and 9, 8 and 19, or 9 and 19 were used to amplify cDNAs from clone 1.1; primers specific to exons 4 and 17 were used for clones 1.2 and 1.9. The resulting fragments were analyzed by sequencing.

**Table 2.** Primers used for quantitative RT-PCR.

| Primer | Sequence |
| --- | --- |
| LRRCC1-1_Fw | CAA CAA GGA TCT TCT CTA GCC CA |
| LRRCC1-1_Rv | AGT TTG GTC GTC TAT GAT TTT GCA |
| LRRCC1-2_Fw | GCA CAA CAA GGA TCT TCT CTA GC |
| LRRCC1-2_Rv | TCG CAG ACA TTC ATT CTC TCT AGA |
| PTCH1_Fw | CCC CTG TAC GAA GTG GAC ACT CTC |
| PTCH1_Rv | AAG GAA GAT CAC CAC TAC CTT GGC T |
| CHMP2A_Fw | ATG GGC ACC ATG AAC AGA CAG |
| CHMP2A_Rv | TCT CCT CTT CAT CTT CCT CAT CAC |
| EMC7_Fw | GTC AGA CTG CCC TAT CCT CTC C |
| EMC7_Rv | CAT GTC AGG ATC ACT TGT GTT GAC |

## Inducible HEK 293 cell lines

LRRCC1 full-length coding sequence was amplified from cDNA clone IMAGE:5272572 (GenBank accession: BC070092.1), corresponding to the longest isoform of LRRCC1 (NM_033402.5), after correction of a frameshift error by PCR mutagenesis. As N- and C-terminal GFP fusions were not targeted to the centrosome, we inserted the GFP tag within the LRRCC1 sequence in disordered regions present between the leucine-rich repeat and coiled-coil domains, either after amino acid 251 or 402. The fusions were cloned into the pCDNA-5FRT (Thermo Fisher Scientific) vector using the Gibson assembly method (*Gibson et al., 2009*) and then integrated into the Flp-In-293 cell line using the Flp-In system (Thermo Fisher Scientific). Expression of the GFP-LRRCC1 fusions was induced by culturing the Flp-In-293 cell lines overnight in medium supplemented with 1 µg/mL doxycycline (Thermo Fisher Scientific).

## RNAi

Ready to use double-stranded siRNA LRRCC1-si1 (target sequence: 5'-AAG GAG AAA GAT GGA GAC GAT-3') (*Muto et al., 2008*), LRRCC1-si2 (target sequence: 5'-TTA GAT GAC CAA ATT CTA CAA-3'), and control siRNA (AllStars Negative Control) were purchased from QIAGEN. siRNAs were delivered into cells using Lipofectamine RNAiMAX diluted in OptiMEM medium (Thermo Fisher Scientific). Cells were fixed after 48 hr and processed for immunofluorescence. For RNAi depletion of ciliated cells, RPE1 cells grown in complete culture medium were treated by RNAi, incubated for 2 days, then submitted to a second round of RNAi. After 8 hr, cells were washed 3× in PBS then cultured during 24 hr in serum-free medium to induce ciliogenesis.

## qRT-PCR

Total RNA extracts were obtained using the Nucleospin RNA kit (Macherey-Nagel) and cDNAs were synthetized using SuperScript III reverse transcriptase (Thermo Fisher Scientific). qPCR was performed in triplicate with the GoTaq qPCR Master Mix (Promega) in a LightCycler 480 instrument (Roche) using the primers listed in *Table 2*. Quantification of relative mRNA levels was performed using CHMP2A and EMC7 as reference genes following the MIQE guidelines (*Bustin et al., 2009*).

## Antibodies

Fragments encoding either aa 671–805 (Ab1) or aa 961–1032 (Ab2) of LRRCC1 (NP_208325.3) were cloned in pGST-Parallel1 and expressed in *Escherichia coli*. The GST-fusion proteins were purified under native conditions using glutathione agarose (Thermo Fisher Scientific), and the LRRCC1 fragments were recovered by Tev protease cleavage and dialyzed before rabbit immunization (Covalab). Antibodies were affinity-purified over the corresponding GST-LRRCC1 fusion bound to Affi-Gel 10 resin (Bio-Rad). Other primary and secondary antibodies used in this study are listed in *Table 1*.

## Western blot

For whole-cell extracts, Flp-In-293 cell lines expressing the GFP-LRRCC1 fusions were induced overnight with doxycycline, collected by centrifugation, and resuspended in Western blot sample buffer prior to incubation at 95°C for 5 min. For immunoprecipitation experiments, doxycycline-induced cells expressing LRRCC1 with a GFP inserted after aa 402 were resuspended in lysis buffer (50 mM Tris pH 8, 150 mM NaCl, 1% NP-40, 0.5% sodium deoxycholate, 0.1% SDS) supplemented with 1 mM $MgCl_2$, 20 µg/mL DNAse I (Roche), and a protease inhibitor cocktail (cOmplete Mini, EDTA-free, Roche). After 30 min on ice, the lysates were centrifuged at 15,000×$g$ for 10 min at 4°C. The supernatants were then incubated with Dynabeads M-280 sheep-anti rabbit magnetic beads (Thermo Fisher Scientific) previously incubated with rabbit anti-IgGs, either anti-GFP or anti-HA tag for the control IP (*Table 1*), and rotated for 3 hr at 4°C. After 3 washes with lysis buffer, immunoprecipitated proteins were recovered by resuspending the beads in sample buffer and heating at 95°C for 5 min. The samples were then run on 4–20% Mini-Protean TGX precast protein gels (Bio-Rad) and transferred onto PVDF membrane using the iBlot 2 blot system (Thermo Fisher Scientific). The membranes were blocked and incubated with antibodies following standard procedures, then visualized using Pierce ECL plus chemiluminescence reagents (Thermo Fisher Scientific) on a ChemiDoc imaging system (Bio-Rad).

## Immunofluorescence

Cells were fixed in cold methanol for 5 min at – 20°C, blocked 10 min with 3% BSA (Sigma-Aldrich) in PBS containing 0.05% Tween-20 (PBST-0.05%), then incubated with primary antibodies diluted in PBST-0.05% containing 3% BSA for 1 hr. After washing 3 × 1 min in PBST-0.05%, cells were incubated 2 hr with secondary antibodies in PBST-0.05% containing 3% BSA and 5 µg/mL Hoechst 33342 (Thermo Fisher Scientific), washed in PBST-0.05% as previously, and mounted using Fluorescence Mounting Medium (Agilent). For staining of primary cilia with anti-acetylated tubulin, cells were incubated 2 hr on ice prior to methanol fixation. For quantification of SMO accumulation within cilia, confluent cells cultured during 24 hr in serum-free medium were supplemented with 200 nM SAG (Sigma) diluted in DMSO, or DMSO alone for 24 hr. Cells were then co-stained for SMO and ARL13B to determine the position of the primary cilium. For all experiments involving induction of ciliogenesis by serum deprivation, we verified that cells were arrested in G0 by immunofluorescence staining of Ki67. To visualize centriolar LRRCC1 and quantify CEP290 centrosomal levels, cells were treated during 1 hr with 5 µM nocodazole prior to fixation. Images were acquired using an Axio Observer Z.1 microscope (Zeiss) equipped with a sCMOS Orca Flash4 LT camera (Hamamatsu) and a ×63 objective (Plan Apo, N.A. 1.4). The structured illumination microscopy (SIM) image was acquired on an ELYRA PS.1 (Zeiss) equipped with an EMCCD iXon 885 camera (Andor) and a ×63 objective (Plan Apo, N.A. 1.4).

## Ultrastructure expansion microscopy

We used the U-ExM protocol described in *Gambarotto et al., 2019* with slight modifications. Cells grown on glass coverslips were incubated in a fresh solution of 1% acrylamide and 0.7% formaldehyde diluted in PBS. After incubating 5 hr to overnight at 37°C, the coverslips were washed with PBS and placed cells down on a drop of 35 µL monomer solution (19.3% sodium acrylate, 10% acrylamide, 0.1% bis-acrylamide in PBS) to which 0.5% TEMED and 0.1% ammonium persulfate were added just before use. The coverslips were incubated 5 min on ice then 1 hr at 37°C, then transferred to denaturation buffer (200 mM SDS, 200 mM NaCl, 50 mM Tris pH9) for 15 min with agitation to detach the gels from the coverslips. The gels were then incubated in denaturation buffer 1.5 hr at 95°C, washed 2 × 30 min in deionized water, then incubated overnight in water at room temperature to allow expansion of the gel. The gels were measured at this step to determine the coefficient of expansion. After 2 × 10 min in PBS, the gels were cut into smaller pieces then incubated 3 hr at 37°C with primary antibodies diluted in saturation buffer (3% BSA, 0.05% Tween-20 in PBS). The gel fragments were then washed 3 × 10 min in PBST-0.1%, incubated 3 hr with secondary antibodies, and washed in PBST-0.1% as described previously. Finally, the gels were incubated 2 × 30 min in deionized water, then left to expand overnight in deionized water to regain their maximum size. For U-ExM of mouse tracheal cells, a fragment of WT mouse trachea (kind gift from I. Le Parco, IJM, Paris) was adhered on a poly-lysine-coated coverslip, then processed as described above with the following modifications: for the first step, the fragment of trachea was incubated overnight to 48 hr in 1% acrylamide and 0.7% formaldehyde in PBS; they were placed 15 min on ice prior to the 1 hr incubation at 37°C and the transfer to

denaturation buffer. Note that GFP fluorescence was quenched during U-ExM processing, so the GFP-Cen2 construct expressed in ependymal cells was not detectable in final samples. Gels were imaged on Lab-Tek chamber slides (0.15 mm) coated with poly-lysine (Thermo Fisher Scientific). Images were acquired at room temperature using either a LSM780 confocal microscope (Zeiss) equipped with an oil ×63 objective (Plan Apo, N.A. 1.4) or an LSM980 confocal microscope with Airyscan 2 (Zeiss) equipped with an oil ×63 objective (Plan Apo, N.A. 1.4).

## Image analysis

Protein levels were determined using ImageJ software (*Schneider et al., 2012*) by measuring the fluorescence intensity in the centrosome or cilium area and subtracting the cytoplasmic background in z-series taken at 0.5 µm interval. Images of individual centrioles in U-ExM are maximum intensity projections of all z-sections comprising the signal of interest. Note that centrioles are presented as they are in the sample (i.e., without correcting their orientation), which leads to an apparent shift between channels or decreased circularity in the projections when centrioles are not parallel to the imaging axis. Analysis of DA morphology defects was performed on z-stacks and not on projected images. Daughter centriole length was determined by U-ExM using the acetylated tubulin staining. For mother centrioles, which could be associated with a primary cilium, the length was measured between the proximal end of the acetylated tubulin staining and DAs labeled by anti-CEP164 or CEP83. To generate average images of LRRCC1 and C2CD3, only centrioles that were nearly perpendicular to the imaging plane were acquired on the Airyscan microscope in order to maximize the resolution in transverse views. Calculating the average image consisted of several steps: cropping out individual centrioles, aligning them, providing reference points, standardizing centrioles using the reference points, and averaging (*Figure 1—figure supplement 2*). The cropping was done in ImageJ, and for aligning and providing the reference points a graphical user interface was developed based on Napari (*Sofroniew et al., 2020*). Centriole alignment: the direction of centriole long axis was selected manually and used to position the centriole vertically. Providing the reference points: reference points were manually selected to outline the circle of microtubules triplets and the location of the protein of interest. The centriole was also framed in Z dimension with a rectangle. Standardization: the reference points were used to calculate all necessary transformations (rotation, scaling, and translation) to map the original image of a centriole to the standard image. Averaging: an average image was calculated for all the successive XY planes of the standardized image stacks. For alignment of tracheal cell centrioles, since the current version of the graphical user interface can only accommodate two channels, the position of the basal foot provided by the γ-tubulin channel was reported manually in the acetylated tubulin channel using ImageJ. The images were then processed as before using the manual annotation as a reference point for the basal foot.

For analysis of procentriole position and LRRCC1 location in procentrioles, 3D reconstructions of diplosomes processed for U-ExM were obtained using Imaris software (Oxford Instruments).

## Electron microscopy

RPE1 cells were grown at confluence before induction of ciliogenesis for 72 hr by serum deprivation. Cells were fixed 30 min in 2.5% glutaraldehyde (Electron Microscopy Sciences), 2% paraformaldehyde (Electron Microscopy Sciences), 1 mM $CaCl_2$ in PBS, then washed 3 × 5 min in PBS. Samples were then post-fixed during 30 min in 1% osmium tetroxide (Electron Microscopy Sciences), then washed 3 × 5 min in water. Dehydration was performed using graded series of ethanol in water for 5 min 30, 50, 70, 90, 100, and 100%. Resin infiltration was performed by incubating 30 min in an Agar low-viscosity resin (Agar Scientific Ltd) and EtOH (1:2) mix, then 30 min in a resin and EtOH (2:1) mix followed by overnight incubation in pure resin. The resin was then changed and the samples further incubated during 1.5 hr prior to inclusion in gelatin capsules and overnight polymerization at 60°C. 70 nm sections were obtained using an EM UC6 ultramicrotome (Leica), post-stained in 4% aqueous uranyl acetate and lead citrate, and observed at 80 kV with a Tecnai12 transmission electron microscope (Thermo Fisher Scientific) equipped with a 1K × 1K Keen View camera (OSIS).

## Videomicroscopy

To determine the duration of mitosis, individual frames of cells growing under normal culture conditions were acquired every 5 min for 24 hr using an IncuCyte ZOOM live-cell analysis system (Sartorius) equipped with a ×20 objective.

## Statistical analysis

All statistical analyses were performed using the Prism 9 for Mac OS X software (GraphPad Software, Inc). All values are provided as mean ± SD. The number of experimental replicates and the statistical test used are indicated in the figure legends, and the p-values are included when statistically different.

## Acknowledgements

We are deeply grateful to Marine Laporte, Virginie Hamel, Paul Guichard, and Davide Gambarotto for teaching us the U-ExM procedure and for sharing antibodies; Arnaud Echard and Takashi Ochi for critical reading of the manuscript; Amélie-Rose Boudjema and Alice Meunier for providing mouse ependymal cells and Isabelle Le Parco for the tracheal tissue; Juliane Da Graça and Simon Herman for technical help; Rémi Le Borgne for help with transmission electron microscopy and for critical reading of the manuscript. We acknowledge the core imaging facility of Institut Jacques Monod (ImagoSeine facility, member of the France BioImaging infrastructure supported by grant ANR-10-INBS-04 from the French National Research Agency). This work was supported by funding from La Ligue Contre le Cancer, Fondation ARC pour la recherche sur le cancer, Labex *Who Am I*? (supported by grants ANR-11-LABX-0071 and ANR-11-IDEX-0005-02) and ANR-21-CE13-008 to JA. NG was a recipient of a MESRI PhD fellowship from the French government and a 4th year PhD fellowship from the Fondation pour la Recherche Médicale.

## Additional information

### Funding

| Funder | Grant reference number | Author |
| --- | --- | --- |
| Agence Nationale de la Recherche | ANR-21-CE13-008 | Juliette Azimzadeh |
| Fondation pour la Recherche Médicale | Graduate Student Fellowship | Noémie Gaudin |
| Fondation ARC pour la Recherche sur le Cancer | ARCPJA32020060002055 | Juliette Azimzadeh |
| Ligue Contre le Cancer | RS16/75-105 | Juliette Azimzadeh |
| Labex Who Am I? | Thematic Program | Juliette Azimzadeh |

The funders had no role in study design, data collection and interpretation, or the decision to submit the work for publication.

### Author contributions

Noémie Gaudin, Conceptualization, Formal analysis, Funding acquisition, Investigation, Methodology, Project administration, Resources, Supervision, Validation, Visualization, Writing – original draft; Paula Martin Gil, Data curation, Formal analysis, Investigation, Supervision, Validation, Visualization, Writing – original draft; Meriem Boumendjel, Manon Bouix, Quentin Delobelle, Lucia Maniscalco, Formal analysis, Investigation, Visualization; Dmitry Ershov, Methodology, Resources, Software, Visualization; Catherine Pioche-Durieu, Investigation, Visualization; Than Bich Ngan Phan, Formal analysis, Investigation, Validation; Vincent Heyer, Bernardo Reina-San-Martin, Resources; Juliette Azimzadeh, Conceptualization, Data curation, Formal analysis, Funding acquisition, Investigation, Project administration, Supervision, Validation, Writing – original draft

### Author ORCIDs

Catherine Pioche-Durieu (ID) http://orcid.org/0000-0003-0988-1169

Juliette Azimzadeh http://orcid.org/0000-0002-7292-9973

**Decision letter and Author response**
Decision letter https://doi.org/10.7554/eLife.72382.sa1
Author response https://doi.org/10.7554/eLife.72382.sa2

## Additional files

### Supplementary files
• Transparent reporting form

### Data availability
All data generated or analyzed during this study are included in the manuscript and supporting files. Source data files are available from the Dryad database (doi:https://doi.org/10.5061/dryad.95x69p8m5).

The following dataset was generated:

| Author(s) | Year | Dataset title | Dataset URL | Database and Identifier |
|---|---|---|---|---|
| Azimzadeh J | 2022 | Data from: Evolutionary conservation of centriole rotational asymmetry in the human centrosome | http://dx.doi.org/10.5061/dryad.95x69p8m5 | Dryad Digital Repository, 10.5061/dryad.95x69p8m5 |

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
