## [Editor Report]

This work shows that, contrary to a widely accepted view, centrioles of the human centrosome are rotationally asymmetric, a feature previously known only from centrioles in flagellated protists and multiciliated cells. The authors identify LRRCC1, implicated in ciliary disease, as an asymmetrically localized protein of the centriole lumen and show that it contributes to proper centriole structure, ciliary assembly, and ciliary signaling.

---

## [Decision Letter]

**Decision letter after peer review:**

Thank you for submitting your article "Evolutionary conservation of centriole rotational asymmetry in the human centrosome" for consideration by *eLife*. Your article has been reviewed by 3 peer reviewers, one of whom is a member of our Board of Reviewing Editors, and the evaluation has been overseen by Anna Akhmanova as the Senior Editor. The reviewers have opted to remain anonymous.

Essential revisions:

1) The most important finding is the demonstration of rotational asymmetry of LRRCC1 localization. The reviewers felt that this finding needs to be confirmed by an independent imaging method. As indicated by the reviewers, there are multiple options, but 3D-SIM may be the most accessible.

2) A careful analysis of centriole numbers in LRRCC1 deficient cells (e.g. at the end of the duplication cycle in mitotic cells) should be performed to determine whether correct duplication occurs.

3) All reviewers had concerns regarding the functional data in the partially depleted KO line. Could the authors use RNAi in the KO cell lines to achieve further depletion and more robust phenotypes? Related to this, the authors should use super-resolution imaging to demonstrate reduced levels at centrioles (not only at centrosomes or pericentriosomal region/satellites) in the KO lines.

*Reviewer #1 (Recommendations for the authors):*

The authors convincingly demonstrate rotational asymmetry in human centrosome centrioles and a contribution of LRRCC1 to cilium assembly and ciliary signaling. Overall the data is of high quality and mostly support the conclusions. There are a few points, however, that need to be addressed:

1) Does LRRCC1 affect centriole assembly? Is there a change in centriole number in the partial KO lines or after RNAi? This is an obvious question when analyzing the function of a centriole component and is also an important control when claiming cilium assembly defects. The authors should quantify centriole number in mitotic cells.

2) Considering the mild phenotypes after partial depletion of LRRCC1, could the authors use RNAi in the KO cell lines to achieve stronger depletion and more robust phenotypes?

3) Figure 5. Changes in centriole length: were these measurements done at specific cell cycle stages? Can the authors exclude that cell cycle effects in cells with reduced LRRCC1 levels may cause differences in measured centriole length? Related: is mitotic progression normal in these cells? Delayed mitotic progression was shown to also cause centriole elongation (e.g. doi: 10.1083/jcb.201910019). Since full knockout seem to prevent cell growth, it is possible that partial KO may also cause proliferation defects. FACS analysis and monitoring the timing of mitotic progression would address this.*Reviewer #2 (Recommendations for the authors):*

The data appear too preliminary for immediate publication in *eLife*. Nevertheless, the subject is intriguing and if authors could provide a cleaner evidence of LRRCC1 and C2CD3 localization, show how depletion of LRRCC1 affects centriole, appendage and cilia structure, the manuscript would be suitable for publication.

Point 1. Expansion microscopy is a powerful imaging approach. However, it is used here to determine localization of a relatively uncharacterized protein, using uncharacterized antibodies. This is risky. For instance: In figure 5 c-e, the authors judge the arrangement of Cep164 to classify centrosomes into 'normal, abnormal, and missing'. This characterization is made based on uneven distribution of Cep164 signals. However, if this criterion is to be extended to acetylated tubulin signal across all expansion images, many of control centrioles (Figure 1c, centriole2 and 3, Figure 3, centriole in b, 1.1 and 1.9 centrioles in Figure 6 f) must also be deemed 'aberrant', due to distortions, uneven spacings between MT blades etc. Clearly, expansion microscopy has its pitfalls.

There are also huge inconsistencies in the pattern and a great variability in the intensity of the LRRCC1 signals between figures and figure panels. For instance, in non-averaged images of expanded centrioles, LRRCC1 can be found in centriole's lumen, in association with MT wall, and in association with variable number of MT blades. In Figure 2b, and 1 b, LLRRCC1 signal is even associated with procentriole's proximal end. How much of the LLRRCC1 signal is indeed specific? The situation seems even more dramatic with C2CD3 data, where some of the signals lie entirely outside the centriole (for instance in 1.9 clone, Figure 6F). Can authors comment on how LRRCC1 and C2CD3 signals can protrude and be localized beyond centriole wall? Previous work (PMID: 30988386, PMID: 29789620) shows C2CD3 to be centered within centriole distal end, and not asymmetric.

It would be critical to use an additional super resolution microscopy technique to re-analyze the localization of LRRCC1 and C2CD3 in non-expanded samples and not to rely solely on expansion microscopy.

Point 2. It is not stated in the manuscript how and whether axial and lateral shifts between different channels were corrected? Was such correction done before image averaging or before image assembly?

Instead of showing averaged data, panels showing more individual examples of horizontally and vertically oriented centrioles would be more instructive. Additional supplemental figures could be assembled for this purpose, at least for the purpose of evaluation.

Point 3. Quantification of LRRCC1 levels in CRISPR clones (Figure 4 a) does not reflect centriole-associated levels of LRRCC1. It reflects the loss of pericentrosomal (satellite) population of LRRCC1, which is the most abundant. Quantification of the centriole bound LCCRR1 needs to be provided instead. Are centrosomal levels of LRRCC1 in 1.1, 1.2, and 1.9 CRISPR clones (Figure 4 a) decreased in comparison to the ones from nocodazole -treated cells (Figure 1A)? In addition, higher resolution images of LRRCC1 in CRISPR clones have not been provided to illustrate where the leftover protein is localized in CRISPR 1.1, 1.2, and 1.9 clones.

How can authors exclude the possibility that loss of the pericentrosomal population of LRRCC1 affects ciliation and ciliary signaling?

Point 4. Between 8 independent experiments conducted in wt RPE-1 cells, the percentage of centrioles with 'abnormal or missing" arrangement of Cep164 varies from ~0% – 40% (Figure 5C). This is a large variability since almost all wt RPE-1 cells can form a cilium (Figure 4b). In my view, this means either that the variability in Cep164 signal is introduced experimentally, or that it is physiological and cannot be used as a measure for ciliation. Indeed, previous super-resolution analysis has demonstrated that, although distal appendage proteins display characteristic pattern around centrioles, their levels on individual appendages can vary under physiological conditions (PMID: 30824690, PMID: 29789620). Therefore, significance of the data from Figure 5 c-e is unclear. Besides, additional appendage proteins (such as Cep83 and SCLT1, for instance) should be analyzed to reach a conclusion about distal appendage organization and number.

Point 5. The authors propose that LRRCC1 cooperates with C2CD3 in organization of distal centriole ends. However, partial reduction in LRRCC1 leads to elongated centrioles only in 1.9 CRISPR clone and not in other two clones, but it is unclear why, as differences between the centrosomal levels of LRRCC1 in CRISPR clones do not seem large enough to explain this difference. Increased centriole length in clone 1.9 could also be non-specific due to clonal variation, and it is unclear whether it is relevant. Especially, because both clones 1.1 and 1.9 seem to show similar defects in ciliation and Cep164 assembly.

To prove and understand what type of centriolar structural defects occur after LRRCC1 depletion, extensive ultrastructural analysis would need to be conducted. Perhaps LRRCC1 could be further depleted by siRNA in LRRCC1- CRISPR clones to generate more penetrant phenotypes?

Point 6. Figure 7. More evidence of LRRCC1/C2CD3 colocalization needs to be provided. Why is majority of C2CD3 signal colocalized with centriole MTs?

*Reviewer #3 (Recommendations for the authors):*

Critical to the central points of the paper are the asymmetric localization of LRRCC1 and C2CD3 at the distal centriole. It is not clear whether the expansion in expansion microscopy is always isotropic and anisotropic expansion could conceivably skew protein localization to appear asymmetric. Could an orthogonal superresolution method, such as 3D-SIM/STED/STORM be used to confirm asymmetry of LRRCC1 or C2CD3? Additionally, could performing 3D-reconstruction image analysis of LRRCC1/C2CD3 with a distal appendage component or another symmetric sub-component of the distal centriole is critical rule out potential artifacts that could be due to anisotropic expansion during U-ExM? The authors imaged OFD1 or CEP290 using U-ExM (fluorescent intensity quantification data are presented in Supplementary Figure S3, but no images are shown). Further confidence in the LRRCC1 conclusions would be garnered by data indicating that these proteins are symmetrically localized at the distal centriole.

The authors report that their results "demonstrate that rotational asymmetry is a conserved ancient property of centrioles." However, the basal foot is a previously described rotational asymmetry present in animal cells, no? Indeed, the basal foot seems to be anticorrelated with the localization of LRRCC1. The authors examine distal appendage formation in some depth. Is basal foot formation or location compromised in LRRCC1 depleted cells?

Quantitation of depletion of LRRCC1 is done by light microscopy. The remaining population, although quantitated, is not characterized spatially with any high degree of resolution. Should one subcellular population (centriolar versus satellite) be preferentially depleted by the reduction in LRRCC1 levels, one population might be implicated in the biological functions of LRRCC1. Where is the residual LRRCC1 staining in the CRISPR/Cas9 generated clones? For example, figure 4a would benefit from U-ExM and quantification of fluorescence intensity specifically at centrioles to determine if the centriolar pool of LRRCC1 is disrupted in the hypomorphs, as surmised by the authors.

It is curious that the authors have been unable to generate null mutations in LRRCC1. RPE1 cells fail to proliferate in the presence of centrinone, possibly hinting at a critical role for LRRCC1 in centriole duplication. Is under duplication or delayed duplication of centrioles observed in LRRCC1 depleted cells?

Relatedly, were null mutations in LRRCC1 generated in HEK 293 cells?

In the discussion, the authors speculate at some length about a role of LRRCC1 in the formation of the ciliary gate, given that LRRCC1 depletion inhibits the ciliary localization of Smoothened. However, one component of the ciliary gate, CEP290, is not reported to be altered in LRRCC1 knockdown. Can the authors examine the dependence of ciliary gate components required for Smoothened localization to cilia in the LRRCC1 knockdown cells?

I recognize that the rescue experiments were difficult. Could an inducible approach (e.g., TetOne system) be attempted? Rescue may help delineate which of the variably penetrant phenotypes are due to downregulation of LRRCC1 and which to other factors. For example, the authors observe a mild increase in centriole length in clone 1.9, but not in clone 1.1. The authors conclude that the slightly lower levels of LRRCC1 in clone 1.9 account for the difference. However, an alternative possibility is that clone 1.9 has another genetic or epigenetic variation that accounts for this incompletely penetrant phenotype.

[Editors' note: further revisions were suggested prior to acceptance, as described below.]

Thank you for resubmitting your work entitled "Evolutionary conservation of centriole rotational asymmetry in the human centrosome" for further consideration by *eLife*. Your revised article has been evaluated by Anna Akhmanova (Senior Editor) and a Reviewing Editor.

The manuscript has been much improved and you have addressed all concerns. There are a couple of items that I would like you to address before formal acceptance:

1) Figure 1d: The Airyscan localization experiment uses POC5 labelling as an example of a symmetrical localization. I suggest that you show the channels also separately so that one can appreciate the luminal POC5 signal better. In the legend you commented on the fact that the POC5 label is also on the outside – in fact the outside label seems stronger than the luminal signal. This is a bit confusing for the purpose of this experiment, but also considering previous POC5 localization studies by ExM (e.g. Le Guennec et al., Sci Adv, 2020; Schweizer et al., Nat Comms, 2021). Is this outside signal specific? It may be good to discuss/explain this a bit more.

2) It is somewhat surprising that you can reliably count individual centrioles using POC5 staining, which is proposed to mark the central lumen. Considering the unexpected distribution of the POC5 label in Fig, 1d, perhaps the more distal, outside label helps? In any case, it would be good to include example cells with POC5 staining of centriole pairs at spindle poles with the quantifications in Figure S1j.

---

## [Author Response]

Essential revisions:1) The most important finding is the demonstration of rotational asymmetry of LRRCC1 localization. The reviewers felt that this finding needs to be confirmed by an independent imaging method. As indicated by the reviewers, there are multiple options, but 3D-SIM may be the most accessible.

In the revised version of the manuscript, we now verify the asymmetry of LRRCC1 localization by a method that does not involve expansion. For this, we imaged conventional immunofluorescence samples using a Zeiss Airyscan 2 confocal microscope. The Airyscan 2 module increases the resolution by a factor of 1.8 compared to conventional confocal microscopy, which is close to the performance of 3D-SIM. The images we obtained are indeed comparable to those obtained using a Zeiss Elyra PS1 microscope (compare the Airyscan images now included in Figure 1d of the revised manuscript with the SIM image in Author response image 1).

**Author response image 1. sa2fig1:** 

To verify the asymmetry of LRRCC1 labeling, we measured the lateral distance between LRRCC1 intensity peak and the long axis of the centriole labeled with acetylated tubulin. To determine centriole orientation with greater accuracy, we measured only mother centrioles associated with a cilium. A comparison between the distribution of LRRCC1 and that of hPOC5, a protein localized symmetrically within centrioles (Le Guennec et al., 2020), confirms the asymmetric localization of LRRCC1 in samples without expansion. These results are now presented in Figure 1d. and discussed in the Results section (p7, lines 122-128).

2) A careful analysis of centriole numbers in LRRCC1 deficient cells (e.g. at the end of the duplication cycle in mitotic cells) should be performed to determine whether correct duplication occurs.

We never observed anomalies in centriole number in cells depleted in LRRCC1, but we had indeed neglected to formally quantify this. We have now included quantification of centriole number in LRRCC1-depleted cells in Supplemental Figure S1j of the revised manuscript. As suggested by the reviewers, we analyzed the number of centrioles at mitotic spindle poles using the hPOC5 protein as a centriole marker. We detected no difference between cells depleted in LRRCC1, either by CRISPR or RNAi, and control cells. We conclude that LRRCC1-depletion does not affect centriole duplication.

3) All reviewers had concerns regarding the functional data in the partially depleted KO line. Could the authors use RNAi in the KO cell lines to achieve further depletion and more robust phenotypes?

As suggested, we treated the CRISPR clones with RNAi to further reduce LRRCC1 levels. Overall LRRCC1 levels in the centrosome area were decreased after RNAi compared to CRISPR clones (Figure 4a of the revised manuscript). We then analyzed several aspects of the LRRCC1 phenotype in CRISPR cells treated with RNAi.

– We first show that further reducing LRRCC1 levels in CRISPR clones does not enhance the ciliogenesis defect, as the proportion of cells forming a primary cilium in these conditions is comparable to what is observed in CRISPR clones treated with a control siRNA. We conclude that depletion of LRRCC1 only partially inhibits ciliogenesis. These results are now presented in Figure 4c of the revised manuscript and discussed in the Results section (p11-12, lines 246-248).

– We show that RNAi treatment leads to a significant increase in mother centriole length in CRISPR clone 1.1 compared to the control. We conclude that depletion of LRRCC1 increases centriole length. These results are now presented in Figure 5b of the revised manuscript and discussed in the Results section (p12-13, lines 271-275).

– We analyzed the effect of RNAi treatment on the recruitment of a second component of distal appendages, CEP83. We show that the proportion of centrioles with abnormal CEP83 labelling is significantly less in RNAi-treated CRISPR clones 1.1 and 1.9 than in control cells, confirming that LRRCC1-depletion leads to abnormalities in distal appendages. These results are now presented in Figure 5g, h of the revised manuscript and discussed in the Results section (p13, lines 284-294).

– We determined how further reducing LRRCC1 levels affects C2CD3 localization pattern. The average images obtained after RNAi treatment of CRISPR clones show that while the treatment does not alter further the C2CD3 pattern in clone 1.9, which has lower levels of centriolar LRRCC1 (see below), it does so in clone 1.1. These results confirm that LRRCC1 depletion affects C2CD3 localization and are now presented in Figure 6g of the revised manuscript (see also Results section, p14-15, lines 321-322). Please note that we generated new average images of the C2CD3 staining for all conditions (Figure 6d and 6g; see the response to point 2 of reviewer #2 for details).

Altogether, our results confirm that LRRCC1 depletion interferes with ciliogenesis, increases centriole length, and leads to abnormal distal appendage morphology and C2CD3 recruitment.

Related to this, the authors should use super-resolution imaging to demonstrate reduced levels at centrioles (not only at centrosomes or pericentriosomal region/satellites) in the KO lines.

In the revised manuscript, we now quantify LRRCC1 levels at the distal end of centrioles by Airyscan microscopy. To identify the distal centriole pool with greater accuracy, we only analyzed mother centrioles associated with a primary cilium. Our results confirmed a decrease in the amounts of centriolar LRRCC1 in all 3 CRISPR clones, with lower levels in clone 1.9, for which we observed the strongest phenotypes. These results are now presented in Figure 4b of the revised manuscript (see also Results section, p11, lines 235-239).

Unexpectedly, the amount of distal protein detected in clone 1.1 was comparatively higher than in the other clones (on the order of 80% of the WT level, compared with < 50% for clone 1.9). Clone 1.1 has deletions in both copies of the *LRRCC1* gene, which is expected to cause a loss of the reading frame in both cases (now in Supplemental Figure S3a of the revised manuscript). To determine what the proteins detected in clone 1.1 correspond to, we performed an analysis of the transcripts present in this clone. As now shown in Supplemental Figure S3b, we identified two long in-frame transcripts. In both cases, an exon adjacent to the exon deleted in the genomic sequence was eliminated by splicing, allowing frame recovery. These splicing profiles are not found in the databases, which supports that the corresponding proteins are mutant isoforms. In contrast, WT transcripts are detected, but at reduced levels compared to WT, in clones 1.2 and 1.9. Together, these analyses show that all 3 clones exhibit reduced levels of centriolar LRRCC1, and that in the case of clone 1.1 only mutant isoforms are present.

Reviewer #1 (Recommendations for the authors):The authors convincingly demonstrate rotational asymmetry in human centrosome centrioles and a contribution of LRRCC1 to cilium assembly and ciliary signaling. Overall the data is of high quality and mostly support the conclusions. There are a few points, however, that need to be addressed:1) Does LRRCC1 affect centriole assembly? Is there a change in centriole number in the partial KO lines or after RNAi? This is an obvious question when analyzing the function of a centriole component and is also an important control when claiming cilium assembly defects. The authors should quantify centriole number in mitotic cells.

Please see the response to point 2 of the Essential Revisions.

2) Considering the mild phenotypes after partial depletion of LRRCC1, could the authors use RNAi in the KO cell lines to achieve stronger depletion and more robust phenotypes?

Please see the response to point 3 of the Essential Revisions.

3) Figure 5. Changes in centriole length: were these measurements done at specific cell cycle stages? Can the authors exclude that cell cycle effects in cells with reduced LRRCC1 levels may cause differences in measured centriole length? Related: is mitotic progression normal in these cells? Delayed mitotic progression was shown to also cause centriole elongation (e.g. doi: 10.1083/jcb.201910019). Since full knockout seem to prevent cell growth, it is possible that partial KO may also cause proliferation defects. FACS analysis and monitoring the timing of mitotic progression would address this.

Centriole length measurements were not taken at a specific time in the cell cycle. However, we measured the length of mother and daughter centrioles separately (Figure 5a) and found both to be increased in clone 1.9. Since the length of the mother centriole does not vary over time, it should not be affected by the cell cycle stage at which it is measured. We have now also included an analysis of centriole length in RNAi-treated clones 1.1 and 1.9 showing that in both cases, mother centriole length is increased compared to the control (Figure 5b; see also point 3 of the Essential Revisions).

As suggested by the reviewer, we have measured the duration of mitosis in CRISPR clones to determine whether centriole elongation might be caused by extended mitosis. As shown in Supplemental Figure S1k of the revised manuscript, we found no difference in mitosis duration between clone 1.9 and WT cells (clones 1.1 and 1.2 show an increase of approximately 3 min on average compared to control cells). We conclude that centriole elongation is not driven by delayed mitosis in clone 1.9.

Reviewer #2 (Recommendations for the authors):The data appear too preliminary for immediate publication in eLife. Nevertheless, the subject is intriguing and if authors could provide a cleaner evidence of LRRCC1 and C2CD3 localization, show how depletion of LRRCC1 affects centriole, appendage and cilia structure, the manuscript would be suitable for publication.Point 1. Expansion microscopy is a powerful imaging approach. However, it is used here to determine localization of a relatively uncharacterized protein, using uncharacterized antibodies. This is risky. For instance: In figure 5 c-e, the authors judge the arrangement of Cep164 to classify centrosomes into 'normal, abnormal, and missing'. This characterization is made based on uneven distribution of Cep164 signals. However, if this criterion is to be extended to acetylated tubulin signal across all expansion images, many of control centrioles (Figure 1c, centriole2 and 3, Figure 3, centriole in b, 1.1 and 1.9 centrioles in Figure 6 f) must also be deemed 'aberrant', due to distortions, uneven spacings between MT blades etc. Clearly, expansion microscopy has its pitfalls.There are also huge inconsistencies in the pattern and a great variability in the intensity of the LRRCC1 signals between figures and figure panels. For instance, in non-averaged images of expanded centrioles, LRRCC1 can be found in centriole's lumen, in association with MT wall, and in association with variable number of MT blades. In Figure 2b, and 1 b, LLRRCC1 signal is even associated with procentriole's proximal end. How much of the LLRRCC1 signal is indeed specific? The situation seems even more dramatic with C2CD3 data, where some of the signals lie entirely outside the centriole (for instance in 1.9 clone, Figure 6F). Can authors comment on how LRRCC1 and C2CD3 signals can protrude and be localized beyond centriole wall? Previous work (PMID: 30988386, PMID: 29789620) shows C2CD3 to be centered within centriole distal end, and not asymmetric.It would be critical to use an additional super resolution microscopy technique to re-analyze the localization of LRRCC1 and C2CD3 in non-expanded samples and not to rely solely on expansion microscopy.

We agree with the reviewer that there are certain points to be careful about when analyzing U-ExM data. Although we systematically inspected our U-ExM samples to ensure that the geometry of the centrioles was preserved, clearly the expansion step can generate additional staining heterogeneity.

We addressed this problem by quantifying our analyses in two ways.

– In the case of distal appendage defects, we categorized centrioles into normal, abnormal and missing DAs, and indeed, we found 'abnormal' centrioles even in control cells. This proportion of abnormal centrioles may result from heterogeneity introduced during sample processing, variability in centriole structure and composition, or a combination of both. Nevertheless, by systematically analyzing multiple series of control and LRRCC1-depleted samples, we found in a highly reproducible manner that distal appendage defects are more frequent in the latter. We have now also added analyses using a second distal appendage marker, CEP83. In this case, it was more difficult to detect morphological variations, as the CEP83 staining is more compact than the CEP164 staining. Nevertheless, we observed a significant decrease in the proportion of centrioles with normal CEP83 labeling after RNAi treatment of CRISPR clones (now in Figure 5g, h). Together, our results support that LRRCC1-depletion affects the morphology of distal appendages.

– For LRRCC1 and C2CD3 labeling, we generated average images to synthesize the information present in the individual images. In our experience, the main issue with U-ExM is that is sometimes lead to suboptimal labeling, even in samples that expanded in an isometric fashion. In contrast, we have no evidence that this method leads to non-specific labeling, unlike SIM for which the reconstruction step can generate artifacts. Generating an average image from individual images is an approach that also has limitations, but which we believe is relevant to our analyses because it allows us to highlight the most salient aspects of LRRCC1 and C2CD3 localization patterns in WT cells. In LRRCC1-depleted cells, in which the variability of the C2CD3 staining is much greater, the average images allow us to visualize a divergence from the control pattern.

Regarding the aberrant patterns of C2CD3 such as the one mentioned by the reviewer, we would like to point out that these were not observed in control samples. Also, the individual centriole images shown in the figures have not been reoriented with respect to the observation axis. The images included in Figures 1c, 5d and g, 6d and f, are maximum intensity projections of individual z-sections encompassing the signal of interest in centrioles as they are in the sample. When the centrioles are tilted, the image projection generates an apparent shift between the two channels, which is due to the fact that the acetylated tubulin staining is fainter in the most distal part of the centriole where the different markers localize (see the lateral view of the average centriole in Figure 1f). This particular point is now addressed in the Material and Methods section (p 28, lines 638-642), as well as in the corresponding figure legends. To address this problem, we included in our image analysis tool a step allowing to reorient the centrioles before generating an average image. Unfortunately, this pipeline in its current state allows us to visually inspect individual centrioles after reorientation, but not to save them (only the average image can be saved). We hope in the future to further develop the potentialities of this tool.

This being said, the channels were not properly aligned in some of the images presented in the previous version of the manuscript. In some cases, it accentuated the shift between channels, including in the image the reviewer is referring to (now Figure 6g, bottom right panel). As explained in more detail in the response to the following comment (Point 2), this problem has now been addressed in the revised version of the manuscript.

Finally, we have now included data showing that LRRCC1 asymmetric localization can be detected in non-expanded samples using Airyscan microscopy (see response to point 3 of the Essential Revisions).

Point 2. It is not stated in the manuscript how and whether axial and lateral shifts between different channels were corrected? Was such correction done before image averaging or before image assembly?Instead of showing averaged data, panels showing more individual examples of horizontally and vertically oriented centrioles would be more instructive. Additional supplemental figures could be assembled for this purpose, at least for the purpose of evaluation.

The alignment of the different channels was verified on series of individual images before generating average images. For this, we took advantage of the fact that the LRRCC1 and C2CD3 antibodies weakly label the centriole wall (see Figure 1b in the case of LRRCC1 labeling), which allows us to verify the overlap with acetylated tubulin labeling. We systematically checked the superposition of the channels in the first and last few images of each series. This was to verify that the U-ExM gel was not drifting, as sometimes happens, and that the microscope was correctly set up for acquisition. However, we have now checked each individual image and found that despite these precautions, the channels were incorrectly aligned in some of our C2CD3 images. No such misalignment was detected in LRRCC1 images. We therefore realigned the C2CD3 images using the weak centriole wall staining generated by the anti-C2CD3 antibody. Images in which such staining was absent or ambiguous were removed and replaced with new images.

In order to remain neutral to the hypothesis of an asymmetric vs. symmetric positioning of C2CD3 within the centriole, we now generated a new average using the shape of the staining pattern to superimpose images of individual centrioles. The resulting image shown in Figure 6e of the revised manuscript confirms the C-shape of C2CD3 labeling but does not show clear positional asymmetry within the centriole. This result is more consistent with previously published results (Yang et al., 2018; Tsai et al., 2019), as pointed above by the reviewer. Interestingly, a filament forming an ‘incomplete circle’ and lining the inner centriole wall was observed in the same region by electron microscopy (Vorobjev and Chentsov, 1980), which may correspond to the C2CD3-pattern we describe (see Discussion section p 17, line 374-376).

Individual images of C2CD3 in CRISPR clones were all reanalyzed and average images were generated in the same manner as for the WT (Figure 6f, g of the revised manuscript). We now also include images of CRISPR cells treated by RNAi to further reduce LRRCC1 levels. These results confirm that LRRCC1 depletion disrupts the C2CD3 localization pattern.

Point 3. Quantification of LRRCC1 levels in CRISPR clones (Figure 4 a) does not reflect centriole-associated levels of LRRCC1. It reflects the loss of pericentrosomal (satellite) population of LRRCC1, which is the most abundant. Quantification of the centriole bound LCCRR1 needs to be provided instead. Are centrosomal levels of LRRCC1 in 1.1, 1.2, and 1.9 CRISPR clones (Figure 4 a) decreased in comparison to the ones from nocodazole -treated cells (Figure 1A)? In addition, higher resolution images of LRRCC1 in CRISPR clones have not been provided to illustrate where the leftover protein is localized in CRISPR 1.1, 1.2, and 1.9 clones.How can authors exclude the possibility that loss of the pericentrosomal population of LRRCC1 affects ciliation and ciliary signaling?

This issue is addressed in the response to point 3 of the Essential Revisions.

Point 4. Between 8 independent experiments conducted in wt RPE-1 cells, the percentage of centrioles with 'abnormal or missing" arrangement of Cep164 varies from ~0% – 40% (Figure 5C). This is a large variability since almost all wt RPE-1 cells can form a cilium (Figure 4b). In my view, this means either that the variability in Cep164 signal is introduced experimentally, or that it is physiological and cannot be used as a measure for ciliation. Indeed, previous super-resolution analysis has demonstrated that, although distal appendage proteins display characteristic pattern around centrioles, their levels on individual appendages can vary under physiological conditions (PMID: 30824690, PMID: 29789620). Therefore, significance of the data from Figure 5 c-e is unclear. Besides, additional appendage proteins (such as Cep83 and SCLT1, for instance) should be analyzed to reach a conclusion about distal appendage organization and number.

This issue is addressed in the response to the reviewer’s point 1.

Point 5. The authors propose that LRRCC1 cooperates with C2CD3 in organization of distal centriole ends. However, partial reduction in LRRCC1 leads to elongated centrioles only in 1.9 CRISPR clone and not in other two clones, but it is unclear why, as differences between the centrosomal levels of LRRCC1 in CRISPR clones do not seem large enough to explain this difference. Increased centriole length in clone 1.9 could also be non-specific due to clonal variation, and it is unclear whether it is relevant. Especially, because both clones 1.1 and 1.9 seem to show similar defects in ciliation and Cep164 assembly.To prove and understand what type of centriolar structural defects occur after LRRCC1 depletion, extensive ultrastructural analysis would need to be conducted. Perhaps LRRCC1 could be further depleted by siRNA in LRRCC1- CRISPR clones to generate more penetrant phenotypes?

This issue is addressed in the response to point 3 of the Essential Revisions.

Point 6. Figure 7. More evidence of LRRCC1/C2CD3 colocalization needs to be provided. Why is majority of C2CD3 signal colocalized with centriole MTs?

We agree with the reviewer that the co-localization data are not entirely satisfactory, but we have tried extensively to get better images using various combinations of antibodies, so far without success. The main difficulty is that far red dyes are not stable under U-ExM conditions – *i.e.* in pure water, which is especially a problem for low intensity signals such as those generated by LRRCC1 or C2CD3 antibodies. This is really an issue we expect to solve in the future. However, although we cannot precisely map the respective locations of LRRCC1 and C2CD3, our images already show that C2CD3 and LRRCC1 localize to the same region of the centriole.

Reviewer #3 (Recommendations for the authors):Critical to the central points of the paper are the asymmetric localization of LRRCC1 and C2CD3 at the distal centriole. It is not clear whether the expansion in expansion microscopy is always isotropic and anisotropic expansion could conceivably skew protein localization to appear asymmetric. Could an orthogonal superresolution method, such as 3D-SIM/STED/STORM be used to confirm asymmetry of LRRCC1 or C2CD3? Additionally, could performing 3D-reconstruction image analysis of LRRCC1/C2CD3 with a distal appendage component or another symmetric sub-component of the distal centriole is critical rule out potential artifacts that could be due to anisotropic expansion during U-ExM? The authors imaged OFD1 or CEP290 using U-ExM (fluorescent intensity quantification data are presented in Supplementary Figure S3, but no images are shown). Further confidence in the LRRCC1 conclusions would be garnered by data indicating that these proteins are symmetrically localized at the distal centriole.

As detailed in the response to point 2 of the Essential Revisions, we have now confirmed that LRRCC1 is asymmetrically localized by a method that does not involve expansion. With respect to C2CD3, as explained in more details in the response to reviewer #2 (point 2), we found that in a subset of C2CD3 images, the two channels were not properly aligned. After systematically correcting this anomaly and generating a new average image based on the shape of the staining pattern, we confirm the C-shape of the C2CD3 pattern but not its asymmetric position in the lumen (Figure 6e and Results section p14, lines 311-316).

Regarding the quantification of OFD1 and CEP290 proteins, it was performed on non-expanded immunofluorescence samples. However, we analyzed CEP164 and now CEP83 localization, and found that these were symmetrically distributed around the centrioles in the majority of WT cells. We made preliminary observations of other markers, including OFD1, CEP290, ODF2 and Talpid3 (see for example the image below, noting that the centriole is tilted, hence the apparent shift between the two channels). We did not observe any obvious asymmetry in these patterns, in sharp contrast to what we observed for LRRCC1. We should also mention that we use the same protocol as in the studies by Le Guennec et al. (2020; doi:10.1126/sciadv.aaz4137), Steib et al. (2020; doi: 10.7554/*eLife*.57205) or Le Borgne et al. (2021, doi.org/10.1101/2021.07.13.452210), which also show symmetric labeling for different markers localized in the centriolar lumen or at distal appendages.

The authors report that their results "demonstrate that rotational asymmetry is a conserved ancient property of centrioles." However, the basal foot is a previously described rotational asymmetry present in animal cells, no? Indeed, the basal foot seems to be anticorrelated with the localization of LRRCC1. The authors examine distal appendage formation in some depth. Is basal foot formation or location compromised in LRRCC1 depleted cells?

Regarding the first point, we meant to say that centriole rotational asymmetry is conserved beyond centrioles that carry asymmetric appendages like in multicellular cells. We have modified this sentence as follows: “Taken together, our results demonstrate that rotational asymmetry is an ancient property of centrioles that is broadly conserved in human cells” (p2, lines 41-43).

Regarding the role of LRRCC1 on basal foot assembly or position, we do not have the expertise for culture and targeted gene inactivation in mammalian multiciliated cells, unfortunately. However, we have shown that in planarian multiciliated cells, inactivation of the LRRCC1 ortholog (SMED-VFL1) indeed leads to defects in basal foot assembly and position (Basquin et al., 2019). Because of this observation, we also tried to determine whether the subdistal appendages of the mother centriole could be impacted at the centrosome of human cells. Unfortunately, subdistal appendages are not properly labelled in WT cells processed for U-ExM with the markers we have tested so far (CEP170, centriolin, ninein). But it is indeed an important point that we want to continue exploring in the future.

Quantitation of depletion of LRRCC1 is done by light microscopy. The remaining population, although quantitated, is not characterized spatially with any high degree of resolution. Should one subcellular population (centriolar versus satellite) be preferentially depleted by the reduction in LRRCC1 levels, one population might be implicated in the biological functions of LRRCC1. Where is the residual LRRCC1 staining in the CRISPR/Cas9 generated clones? For example, figure 4a would benefit from U-ExM and quantification of fluorescence intensity specifically at centrioles to determine if the centriolar pool of LRRCC1 is disrupted in the hypomorphs, as surmised by the authors.

This is addressed in the response to point 3 of the Essential Revisions.

It is curious that the authors have been unable to generate null mutations in LRRCC1. RPE1 cells fail to proliferate in the presence of centrinone, possibly hinting at a critical role for LRRCC1 in centriole duplication. Is under duplication or delayed duplication of centrioles observed in LRRCC1 depleted cells?Relatedly, were null mutations in LRRCC1 generated in HEK 293 cells?

We have now included data showing that centriole duplication is not affected in LRRCC1-depleted cells (Supplemental Figure S1j; see also response to point 2 of the Essential Revisions).

With respect to the lack of LRRCC1 null clones, we also initially thought that this might be related to the fact that RPE1 cells arrest or exhibit slower growth in a p53-dependent manner after a range of centrosome perturbations. However, we also failed to isolate null clones for HEK 293, U2-OS, or p53^-/-^ RPE1 cells (generously donated by Brian Tsou). Thus, we have no explanation at this stage for the fact that null clones have not been isolated, while we have managed to isolate such clones for other genes without difficulty (e.g. CCDC61 and C-Nap1; doi: 10.1111/boc.201900038).

In the discussion, the authors speculate at some length about a role of LRRCC1 in the formation of the ciliary gate, given that LRRCC1 depletion inhibits the ciliary localization of Smoothened. However, one component of the ciliary gate, CEP290, is not reported to be altered in LRRCC1 knockdown. Can the authors examine the dependence of ciliary gate components required for Smoothened localization to cilia in the LRRCC1 knockdown cells?

We agree with the reviewer that this is an important point, but so far we have not been successful with the different antibodies against ciliary gate proteins we tested. We expect to find the right markers and conditions for future studies.

I recognize that the rescue experiments were difficult. Could an inducible approach (e.g., TetOne system) be attempted? Rescue may help delineate which of the variably penetrant phenotypes are due to downregulation of LRRCC1 and which to other factors. For example, the authors observe a mild increase in centriole length in clone 1.9, but not in clone 1.1. The authors conclude that the slightly lower levels of LRRCC1 in clone 1.9 account for the difference. However, an alternative possibility is that clone 1.9 has another genetic or epigenetic variation that accounts for this incompletely penetrant phenotype.

We have tried different approaches for performing rescue experiments, including attempts to generate stable lines expressing inducible constructs, without success. One difficulty we have faced is that the presence of larger tags such as GFP disrupts LRRCC1 localization. Fusions at the N- or C-terminus do not localize at all to the centrosome or the satellites, and fusions with a GFP inserted within the protein sequence (after aa. 251 or 402) localize to the centrosome area and satellites, but we did not detect them at centriole distal ends. We therefore attempted to establish stable lines using a myc-tagged construct since the myc tag is much smaller. Screening non-fluorescent cells is extremely tedious when the proportion of stably transformed cells is low however, and we abandoned this approach after several unsuccessful attempts.

Following the reviewer's recommendation, we have now included data showing the effect of further reducing LRRCC1 levels by RNAi. As also detailed in the response to point 3 of the Essential Revisions, under these conditions we observed a significant difference between clone 1.1 and control cells with respect to centriole length.

[Editors' note: further revisions were suggested prior to acceptance, as described below.]

The manuscript has been much improved and you have addressed all concerns. There are a couple of items that I would like you to address before formal acceptance:1) Figure 1d: The Airyscan localization experiment uses POC5 labelling as an example of a symmetrical localization. I suggest that you show the channels also separately so that one can appreciate the luminal POC5 signal better. In the legend you commented on the fact that the POC5 label is also on the outside – in fact the outside label seems stronger than the luminal signal. This is a bit confusing for the purpose of this experiment, but also considering previous POC5 localization studies by ExM (e.g. Le Guennec et al., Sci Adv, 2020; Schweizer et al., Nat Comms, 2021). Is this outside signal specific? It may be good to discuss/explain this a bit more.

We have now included images showing the individual channels in Figure 1d. We also added a comment in the corresponding legend to specify that the anti-hPOC5 antibody labels a region of the mother centriole that broadly corresponds to the region of distal/subdistal appendages (highlighted in green).

Initially, we considered using OFD1, which localizes distally and near the triplets, as a marker. However, OFD1 labeling is wider and tends to separate into 2 peaks when observed by Airyscan microscopy, which made comparison with LRRCC1 difficult. Similar results were obtained using antibodies against ODF2 and Talpid-3. We therefore opted for hPOC5, which localizes symmetrically in the centriole lumen (Le Guennec et al; 2020; and Schweizer et al., 2021). The antibody we used was described in the initial characterization of the hPOC5 protein, the specificity of which was established via RNAi experiments (Azimzadeh et al., 2009; doi: 10.1083/jcb.200808082). We noted at the time that the immunofluorescence labeling was stronger at the mother centriole than at the daughter centriole, which we interpreted as resulting from a progressive recruitment of hPOC5 into the centriole lumen. The pattern observed by Airyscan microscopy rather suggests that hPOC5 localizes near mother centriole appendages in addition to being present in the centriole lumen. In U-ExM, our antibody only labels the centriole wall however, very similar to what was described by Le Guennec et al. and Schweizer et al. As mentioned in the response to Reviewer #3, none of the antibodies against subdistal appendage components that we tested worked, suggesting that these appendages may not be properly preserved by the U-ExM protocol. If hPOC5 localizes near subdistal appendages, this could explain why this staining is not visible by U-ExM.

In any case, by Airyscan microscopy, hPOC5 localizes as expected in the middle region of centrioles, which is the region in which we performed our measurements. In this region, hPOC5 is clearly localized more symmetrically than LRRCC1, confirming our finding that LRRCC1 localizes asymmetrically in the centriole lumen.

2) It is somewhat surprising that you can reliably count individual centrioles using POC5 staining, which is proposed to mark the central lumen. Considering the unexpected distribution of the POC5 label in Fig, 1d, perhaps the more distal, outside label helps? In any case, it would be good to include example cells with POC5 staining of centriole pairs at spindle poles with the quantifications in Figure S1j.

In the previous version of the manuscript, we forgot to indicate that centrioles were labeled for centrin in addition to hPOC5. An image has now been included in Figure 1—figure supplement 1j, and the figure legend has been modified accordingly (as well as the antibody list in Table 1; highlighted in green). As also shown in our previous study (Azimzadeh et al., 2009), hPOC5 labeling allows to visualize individual centrioles in mitotic cells in a manner similar than centrin.